# Detailed Description and Morphological Assessment of *Sepia typica* (Steenstrup, 1875) (Cephalopoda: Sepiidae)

Robin W. Leslie [1], Anthony J. Richardson [2,3] and Marek R. Lipiński [4,5,*]

1 Department of Forestry, Fisheries and Environment (DFFE), Fisheries Management Branch, Private Bag X2, Vlaeberg, Cape Town 8012, South Africa

2 School of Mathematics and Physics, The University of Queensland, St. Lucia 4072, Australia

3 Commonwealth Scientific and Industrial Research Organization (CSIRO) Oceans and Atmosphere, Queensland Biosciences Precinct (QBP), St. Lucia 4067, Australia

4 Department of Ichthyology and Fisheries Science (DIFS), Rhodes University, P.O. Box 94, Makhanda 6140, South Africa

5 South African Institute of Aquatic Biodiversity (SAIAB), Somerset Rd, Makhanda 6140, South Africa

* Correspondence: lipinski@mweb.co.za

**Abstract:** A detailed systematic account of *Sepia (Hemisepius) typica*, an endemic southern African species of cuttlefish, is presented. An analysis of morphological data (morphometric and meristic characters) suggests that *S. typica* is a single well-established species without morphs or subspecies. It is, however, highly variable, perhaps more so than other small sepiids from the region, and there are slight, but significant indications of population structure. Therefore, molecular biological studies based upon a large sample could help investigate broad genetic patterns in what morphologically appears to be a single highly variable species.

**Keywords:** small cuttlefish; detailed redescription; South African waters; *Hemisepius*





## 1. Introduction

*Sepia (Hemisepius) typica* (Steenstrup, 1875) is easily recognized by the presence of marginal antero-ventral pores on top of distinct glands of as yet unknown function. It is hypothesized that these glands produce a kind of "glue" fastening the animal to the substrate. This was the first small sepiid described from southern Africa in the late 19th century. More than 50 years later, a second small species, *S. robsoni* (MASSY, 1927) was described. A further six species were described over the next century in two brief spates of activity: three in the 19-year period between 1966 and 1985; and three in the 3-year period between 2018 and 2020. Seven of these small sepiid species are relatively well defined and characterized [1–4].

The eighth species, *Sepia typica*, is not well defined because of uncertainty over the significance of morphological differences observed between specimens from the western parts of its range, and the few known specimens from the eastern limits of its wide distribution that were available to previous authors [5–8]. Chun [5] reported two individuals from St. Francis Bay (34°09′ S 24°59′ E, "shallow water"), one of them the first known mature male of *S. typica*. Despite the limited number of specimens described at that time, Thore [6] (p. 50) stated: "I think we have to postulate a constant difference in size between the eastern and western form of *Hemisepius*, the latter being the largest." Thus, Thore [6] assumed that "eastern and western forms" were different and proposed that the eastern forms be named *Hemisepius typicus* var. *chuni*. Adam and Rees [7] expressed concern that the "eastern form" was in fact based on the brief and incomplete description of only one male individual. Roeleveld [8], with much more material from the western region (27 males and 44 females) [8] (pp. 310–311), reviewed Thore's hypothesis by comparing her specimens with data from the literature for the only known specimens from the eastern

region (two males and one female). Her comparison [8] (pp. 262–263), revealed substantial individual variation within the western region with no obvious patterns. All except one of the characters recorded by Chun [5] for the St Francis Bay specimen were encompassed in the variation that she reported for the western region, and she concluded as follows: "Thus Chun's specimen differs from those from more westerly localities only in that it has fewer suckers on the right arm IV, of which none are enlarged distally. Whether or not this constitutes a valid character for separating eastern and western forms of *S. typica* cannot be decided on the basis of the presently known specimens, and a decision must await the collection of further specimens from the eastern coast of South Africa" [8] (p. 264).

The aim of our study was to address these concerns and assess the taxonomic status of the species, based on an examination of 141 new specimens.

## 2. Material and Methods

### 2.1. Data Collection

A large number of specimens of *Sepia typica* were collected by bottom trawl during demersal research surveys conducted off South Africa by the research vessels RS *Africana* and R/V *Dr Fridjof Nansen* between 1987 and 2017. Specimens were fixed in 4–10% buffered formalin and transferred to water first, and then to 70% ethyl alcohol.

From the available material we selected a total of 127 (♂68, ♀59) specimens from three geographical areas for subsequent morphometric and meristic analysis. The three areas (Figure 1) were: a northwest area, offshore off Hondeklip Bay (n = 40; ♂22, ♀18); a central area, Agulhas Bank off Cape Agulhas (n = 38; ♂23, ♀15); and an eastern area, St. Francis and Algoa Bays (inshore) (n = 49; ♂23, ♀26) (Figures 1 and 2). An additional 14 specimens (♂7, ♀7) were selected from the remaining material for dissection, or to illustrate specific characters. The type specimen was seen and examined by the last author in 1975 but these notes were subsequently lost. Sending the type by post or courier between Denmark and South Africa is somewhat risky and was not attempted.

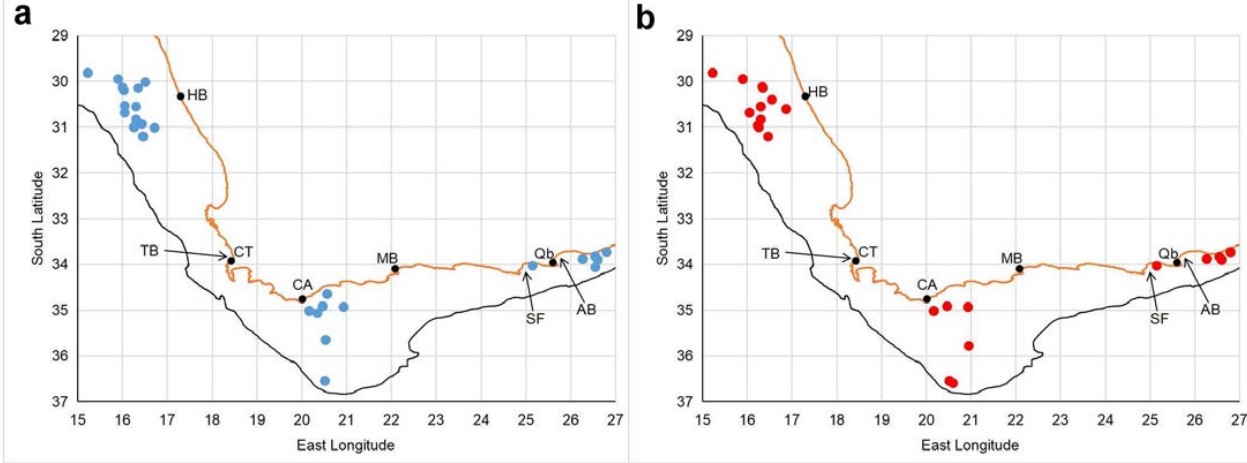

**Figure 1.** Collection localities for male (**a**) and female (**b**) *Sepia typica* specimens analysed for morphological variation. AB = Algoa Bay, CA = Cape Agulhas, CT = Cape Town, HB = Hondeklip Bay, MB = Mossel Bay, Qb = Gqeberha (formerly Port Elizabeth), SF = St Francis Bay, TB = Table Bay (type locality). Dots represent stations where specimens were collected. Orange line represents coastline, and black line the 500 m isobath.

One of the main aims of the study was to assess the significance of the reported morphological variation [6,8] between western and eastern parts of the range of *S. typica* by examining morphological, morphometric, and meristic differences within and between these three areas. St. Francis and Algoa Bays were specifically chosen as our eastern study area to encompass the locality of Chun's [5] specimens.

Meristic and morphometric data collected from all specimens are available as Supplementary Data.

Morphometric measurements and meristic counts used in this study and their acronyms are listed in Table A1. Most of these measurements and counts are defined in the literature [2–4] (available at https://www.foliamalacologica.com) and are not repeated here. However, we have defined a number of new measurements and counts specific to *S. typica* (indicated by an asterisk in Table A1). Usually only the total number of suckers on the left member of each arm pair is recorded, although in many species (including *S. typica*) the suckers are differentiated into fields (See Description, below). We recorded the number of suckers per field. The tip sucker field, distal to the main arm suckers, is distinguished by an abrupt decrease of ~50% in sucker diameter to the first sucker pair in the tip field, with sucker diameter decreasing gradually thereafter towards the tip. There are three counts that are still open to subjectivity: (a) the number of suckers in a transverse row across the middle of the club (ClCR); (b) the number of oblique transverse rows of suckers (TrRC); and (c) the total number of suckers on respective arms—the first two because sucker rows may be strongly oblique or almost perpendicular to the main axis, the last because minute distal suckers fall off easily. These factors may explain differences between our and published readings. Better definitions and improved applications are necessary.

Measurements were taken from preserved specimens. Dorsal (ML) and ventral (MLv) mantle length was measured using digital slide callipers. Fin length (FL) was measured by placing a thread along the base of the fin, marking the length of the fin and then measuring the thread on a metal rule. All other measurements were taken using dividers or a graticule in a stereo dissecting microscope at $10\times$ or $40\times$ magnification. Sucker diameters, anterior dorsal mantle projection (AMH), anterior fin insertion (FIa), and distance between the anterior ventral mantle margin and the first antero-lateral pore on the left and right sides (P1-lt and P1-rt) were measured to the nearest 0.1 mm; fin width (FW) was measured to the nearest 0.5 mm. All other measurements were made to the nearest mm. The range, mean, and standard deviation of the mean and sample size by area and areas combined for each morphometric and meristic character are presented in Tables A2 and A4, respectively, for males and Tables A3 and A5, respectively, for females. All individuals used for the morphological analyses were adults, maturity III–V [9].

Most of the photographs were taken using Canon EOS 7D Mk I and Mk ii cameras, or a Canon EOS 650 camera coupled with a Nikon stereomicroscope using a specially engineered ring. Beaks were photographed using Nikon SMZ18 stereomicroscope with a P2-SHR Apolx lens and NIS Elements D 4.60.00 (build1171) 64bit software.

### 2.2. Data Preprocessing

The following variables were excluded from the statistical analyses

P#-rt, PD-rt, and PR-rt: Pore Count (P#), Pore Distance (PD), and Pore Row length (PR) were recorded for both the left and right side for each specimen. For each of these characters, the count or measurement from the left side (supplemented by the value from the right when the left side was damaged) was retained and the data for the right side were excluded.

ASC1t–ASC4t and HSCt: The number of suckers in the arm-tip field is highly variable (range 3–23, Tables A2–A5). Arm tips are vulnerable and this high variability could be influenced more by injury and sucker loss than by genetic differences between individuals.

TL: The tentacles are elastic and can be retracted into pockets when not in use; consequently, tentacle length (TL) is strongly influenced by condition and whether or not part of the tentacle is retracted into the pockets at the time of preservation.

ClRC: Club row count (ClRC) was invariant (4 suckers per row (Roeleveld [8] counted 6, possibly through taking a more oblique row)).

As all cells in the data matrices must have values for the multivariate analyses; specimens with missing values must either be excluded or the missing value must be estimated. A total of 42% and 33% of our specimens had missing data for one or more morphometric

or meristic character, respectively. Therefore, we screened our data by first removing all specimens that had missing values for more than seven characters, then those characters that had missing values for more than seven individuals. Remaining missing values were replaced by the mean for that variable. For females, 2.0% of meristic and 1.3% of morphometric values were replaced, while for males 2.7% of meristic and 1.1% of morphometric values were replaced.

To remove the effects of size in the morphometric data, we used the residuals from univariate regressions for each character (Table A6) [10,11]. Meristic counts and morphometric residuals were normalised by subtracting the mean and dividing by the standard deviation for each variable to give equal weighting to each character.

### 2.3. Data Analysis

To investigate whether *Sepia typica* differed morphologically amongst our three sampling localities (C = Central, E = Eastern, N = Northwest; Figures 1 and 2), we performed non-metric multidimensional scaling (nMDS). We used the metaMDS function with the Euclidean distance metric in the R package vegan [12]. We performed a separate analysis on meristic counts and for residuals of morphometric measurements separately for males and females.

To test whether *Sepia typica* differed amongst areas, we performed an Adonis (permutational multivariate analysis of variance) analysis with 999 permutations [13]. We then conducted post hoc pairwise comparisons in Adonis [14] to test whether differences between pairs of areas were of significance.

We determined the characters that were primarily responsible for differences amongst areas using an indicator value analysis [15]. This analysis calculates the indicator value of variables as the product of its relative frequency and relative mean abundance in clusters.

In most males, suckers proximal to the tip field on arms I–III p. are differentiated into two regions indicated by a marked increase in sucker diameter, but in some males the suckers are undifferentiated (subequal in diameter) as in females.

To test whether the possession of differentiated or undifferentiated arm suckers represented cryptic polymorphism, or could explain the observed heterogeneity among areas, we used the ratio of the diameter of the smallest proximal sucker (AS1ps) to that of the largest distal sucker (AS1c) of left arm I to divide males into those with differentiated arm suckers and with undifferentiated arm suckers. We performed the same analyses on male meristic counts and residual morphometric measurements as described above, but with specimens grouped by whether they had differentiated (group A) or undifferentiated (group B) suckers on arm I p.

### 3. Results

### 3.1. Geographic Variation in Meristic Counts

All individuals in the plot of nMDS 1 against vMDS 2 for females (Figure 2a) formed a single mixed group with the 95% confidence intervals around the centroids for the three intersecting areas. The permutational multivariate analysis of variance (Adonis) test for differences among areas found no overall heterogeneity ($p = 0.06$). Indicator value analysis found that none of the characters were significant indicators for any of the areas.

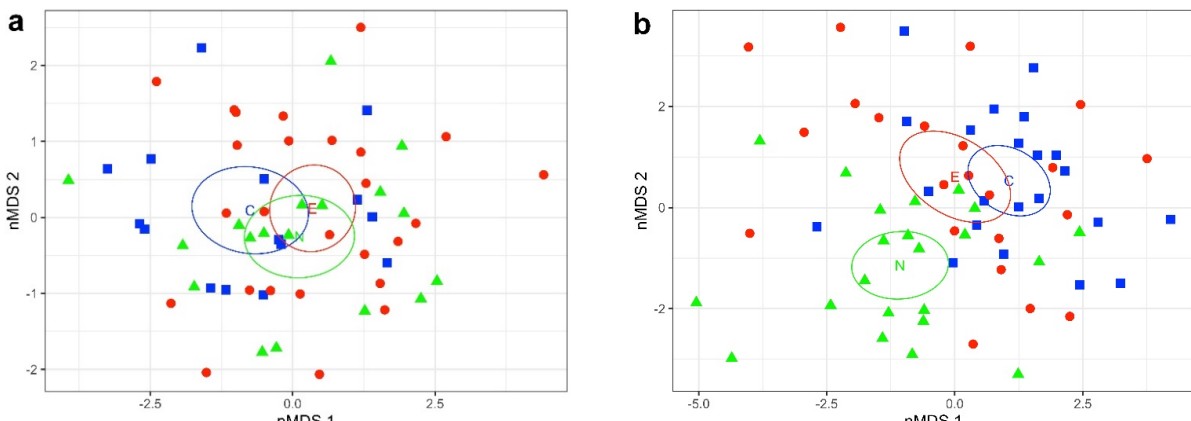

**Figure 2.** *Sepia typica* nMDS plots of meristic counts by area for (**a**) females and (**b**) males. Symbols denote sampling area: Northwest (**green triangles**); Central (**blue squares**); Eastern (**red dots**). Alpha characters denote the centre of distribution per area (C = Central, E = Eastern, and N = Northwest Areas) and the ellipses indicate the 95% confidence interval about the centroids.

In contrast the nMDS for males (Figure 2b) showed the Northwest Area was clearly separated from the Central and Eastern Areas, which were not separated from each other. The Adonis permutation test for differences among areas found highly significant heterogeneity among areas ($p = 0.001$). Pairwise comparisons of areas in Adonis showed that the Northwest Area is highly significantly different from both the Central ($p < 0.001$) and Eastern ($p < 0.001$) Areas, and that the Central and Eastern Areas do not differ significantly from each other ($p = 0.14$). Indicator value analysis showed that ASC1p, ASC3p, and ASC4m ($p < 0.001$) and ASC1p ($p < 0.01$) were significant indicators for the Northwest Area. The number of pores (P#-lt) is a significant ($p < 0.05$) indicator to differentiate between Central and Eastern Areas.

### 3.2. Geographic Variation in Morphometric Measurements

The plot of nMDS 1 against vMDS 2 for females (Figure 3a) showed substantial intersection between the 95% confidence intervals for the Central and Eastern Areas, minimal intersection between the Northern and Central Areas and complete separation of the Northern and Eastern Areas. The permutational multivariate analysis of variance (Adonis) test for differences among areas found highly significant overall heterogeneity ($p < 0.001$). Pairwise comparisons of areas in Adonis showed that the Northwest Area is highly significantly different from both the Central ($p = 0.001$) and Eastern ($p = 0.001$) Areas, and that the Central and Eastern Areas do not differ significantly from each other ($p = 0.45$). Indicator value analysis showed that HL, HW, and AL 4 ($p < 0.001$) and MLv, AS2pl, and AS3pl ($p < 0.01$) are significant indicators for the Northwest Area. The fin length (FL) is a significant ($p < 0.05$) indicator differentiating between the Central and Eastern Areas.

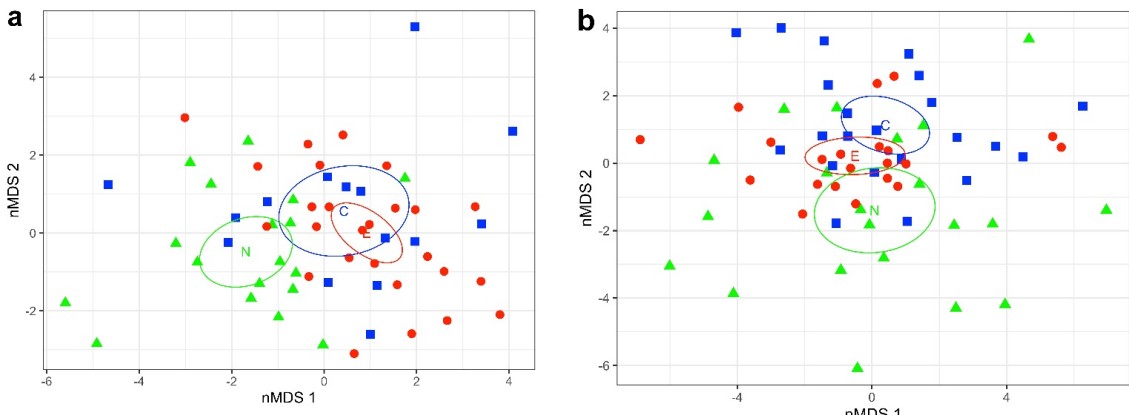

**Figure 3.** *Sepia typica* nMDS plots of morphometric measurement residuals by area for (**a**) females and (**b**) male. Symbols denote sampling area: Northwest (**green triangles**); Central (**blue squares**); Eastern (**red dots**). Alpha characters denote the centre of distribution per area (C = Central, E = Eastern, and N = Northwest Areas) and the ellipses indicate the 95% confidence interval about the centroids.

For males, the nMDS (Figure 3b) showed clear separation between the Northwest and Central Areas, with the Eastern Area intermediate and partially intersecting with both Northwest and Central Areas. The Adonis permutation test for differences among areas found significant heterogeneity among areas ($p < 0.01$). Pairwise comparisons of areas in Adonis showed that the Northwest Area is significantly different from the Central Area ($p < 0.01$). The Eastern Area was not significantly different from either the Northern or Central Areas. Indicator value analysis found that AS1ps, AS1c, AS2c, and AS4pl were significant ($p < 0.01$) indicators for the Northwest Area.

### 3.3. Geographic Variation in Male Arm Morphology

The diameter of the smallest proximal sucker expressed as a proportion of the diameter of the largest distal sucker of left arm I (AS1ps/AS1c) plotted against ML (Figure 4) by area shows that the proportion was not related to size. We chose an AS1ps/AS1c value of 0.8 (indicated by a red line in Figure 4) to divide males into those with differentiated arm suckers (Group A: AS1ps/AS1c < 0.8) and with undifferentiated arm suckers (Group B: AS1ps/AS1c > 0.8). Males with AS1ps/AS1c = 0.8 were regarded as indeterminate and excluded. As all specimens with ML < 16 mm had undifferentiated suckers; it is possible that sucker differentiation only manifests at ML > 16 mm. Therefore, males with ML ≤ 16 mm were excluded forfrom the analysis.

Males with undifferentiated arm suckers were more common in our samples from the Eastern (67%) than from the Northwest (37%) or Central (27%) Areas. The nMDS plots for both meristic counts and residual morphometric measurements showed substantial overlap between the 95% confidence intervals of the two centroids. The permutational multivariate analysis of variance (Adonis) test for differences among areas found no significant overall heterogeneity ($p > 0.1$).

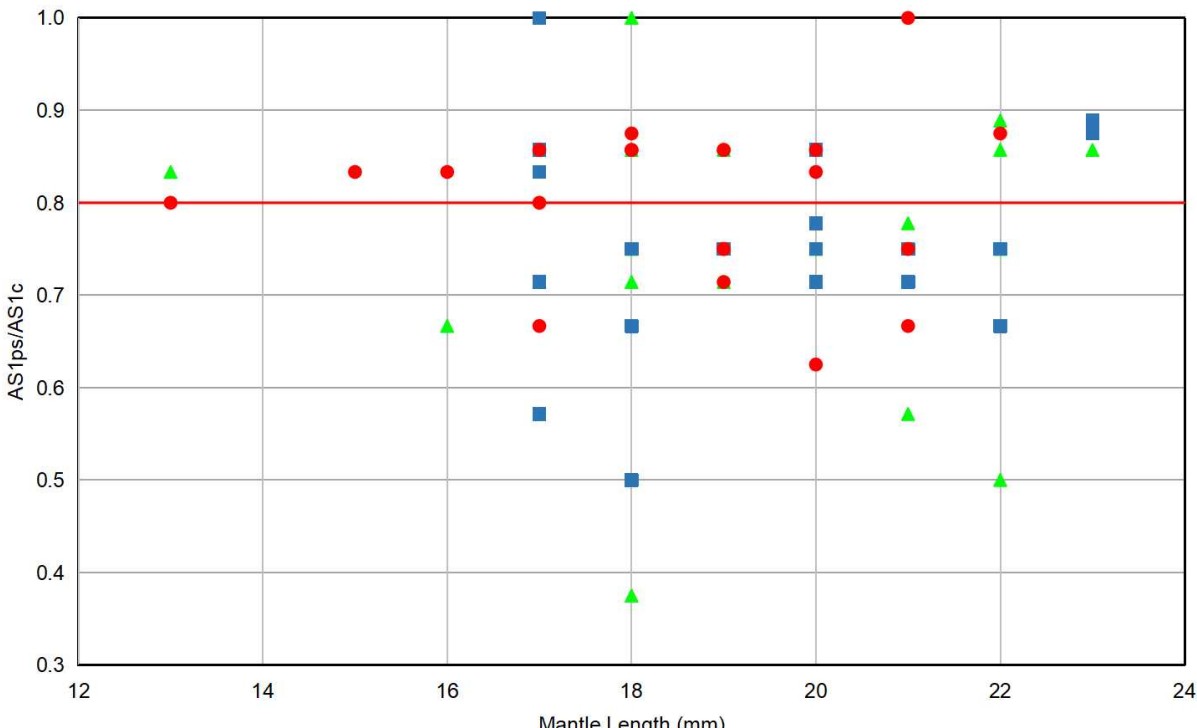

**Figure 4.** The diameter of the smallest proximal sucker as a proportion of the largest distal sucker of arm I p. (AS1ps/AS1c) in male *Sepia typica* plotted against dorsal mantle length (ML; mm). Symbols denote sampling area: Northwest (**green triangles**); Central (**blue squares**); and Eastern (**red dots**). The red line (proportion = 0.8) divides males with differentiated arm suckers (below the line) from those with undifferentiated arm suckers (above the line). Males with proportion = 0.8 are indeterminate.

## 4. Discussion

All four of the nMDS plots presented above (Figures 2 and 3) showed a single loose cluster of data points, with individuals from the three geographic areas mixed to a greater or lesser degree. The 95% confidence intervals around the centroids for the Central and Eastern Areas intersected in all four plots, whereas the Northwest Area showed little or no intersection with the other areas except for the analysis of meristic counts for females (Figure 2a). These visual differences were supported by the pairwise comparisons of areas in Adonis, which found significant differences between the Northwest and Central Areas for three of the analyses and between Northwest and Eastern Areas for two of the four analyses. The Central and Eastern Areas did not differ significantly in any of the analyses.

Indicator value analysis identified ASC1p, ASC3p, ASC4m, and ASC1p as indicators for the Northwest Area for male meristic counts. Student's t-test comparison of means (Table A4) showed that these counts are significantly higher in the Northwest Area than in either the Central or Eastern Areas.

Unfortunately, results of indicator value analyses for morphometric measurement residuals are not easily interpreted, because none of the characters identified as indicators show significant differences among areas either as proportions of ML (Tables A2 and A3) or as regression residuals.

These findings indicate slight, but significant differences among the three sampling sites, but it is unknown whether these differences are site-specific as in possible ecotypes, or the result of different proportions of two cryptic forms at the three sampling sites. A large-scale population genetic study is required to resolve this question.

## 5. Systematic Account

*Sepia typica* (Steenstrup, 1875)

Figures 3–24 and Figures A1–A3; Appendices A–C

*Hemisepius typicus* Steenstrup, 1875: 468 [16]; Hoyle, 1886: 26 and 217 [17], 1912: 281 [18]; Smith, 1903: 356 [19], 1916: 25 [20]; Chun, 1915: 411 [5]; Massy, 1927:164 [21]; Thore, 1945: 50 [6]; Voss, 1962: 248 and 252 [22], 1967: 64 [23].

*Hemisepion typicum* Rochebrune, 1884: 78 [24].

*Rhombosepion* sp. A Massy, 1927: 161 [21].

*Hemisepius typicus* var *chuni* Thore, 1945: 50 [6]; Roeleveld, 1975: 242 [25].

*Sepia typica* Adam and Rees, 1966: 117 [7]; Roeleveld, 1972: 257–264 [8], 1998: 4 [26]; Khromov, 1998: 146 and 155 [27]; Khromov, Lu, Guerra, Dong and Boletzky, 1998: 129 [28]; Reid, Jereb and Roper, 2005: 151 [29]; Leslie and Lipinski, 2018: 347 [30].

**Material Examined**: 141 specimens (75 male and 66 female); see Appendix B.

**Diagnosis**: Dorsal shield of the cuttlebone transparent, chitinous; only phragmocone calcified; striae borderline concave, last (anterior) septum more strongly calcified than the others; other striae simple, horizontal; outer cone broad, transparent. A prominent papilla dorsally on head, above each eye; dorsal mantle structures variable, there is one large tubercle or papilla anteriorly, two in the centre, and usually two smaller tubercles or papillae posteriorly; these main (large) tubercles usually single, but are sometimes tubercle clusters or simple turrets; skin between them usually smooth, but sometimes small tubercles (dispersed and randomly distributed) are present. Two longitudinal rows of glands sub-marginally on antero-ventral mantle (one row on each side), each gland with a dark or light pore, occasionally anterior glands with two pores or posterior glands without pores; usually 10–13 (range 5–15; Ref. [8]; our data 8–15) pores on each side, left and right rows usually with unequal numbers of pores.

**Description:** Small species; mean ± SD ML males 19.0 ± 2.6 mm (Table A2), females 19.1 ± 2.8 mm (Table A3), largest recorded specimen female ML 29 mm (SAIAB 211525). Mantle globose, rounded, squat, but occasionally elongated. Dorso-anterior margin most often a wide W-shape, but rarely simply almost straight or a wide Λ-shape (Figure 5). Ventro-anterior margin variable, from slightly and broadly emarginated to deeply and distinctly emarginated (Figure 6).

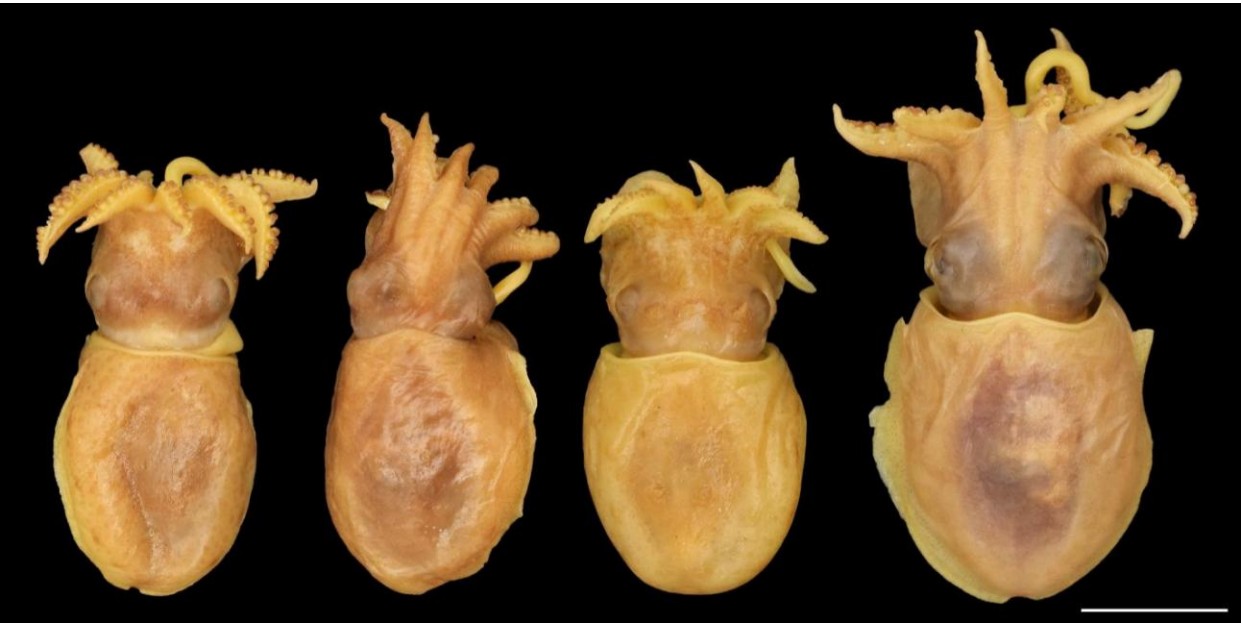

**Figure 5.** Variation in the shape of the dorsal mantle margin in *Sepia typica*: from left to right: SAIAB 211543, male ML 18 mm; SAIAB 211610, male ML 20 mm; SAIAB 211550, female ML 17 mm; SAIAB 211529, male ML 23 mm. Scale bar 10 mm.

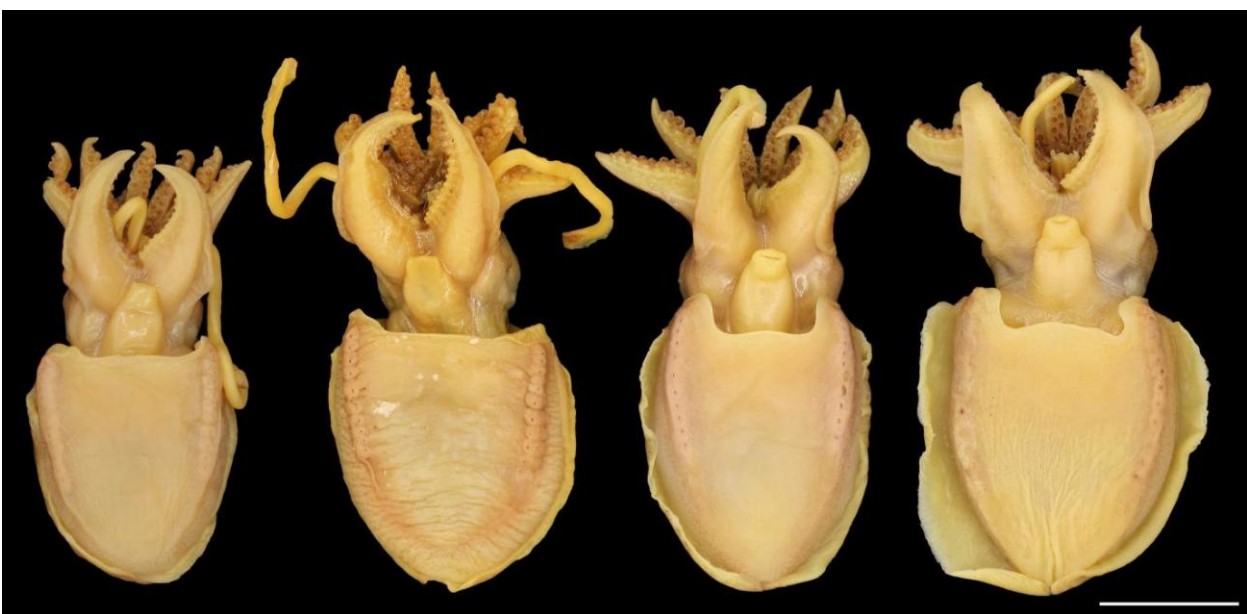

**Figure 6.** Variation in the ventral mantle in *Sepia typica*, especially the degree of emargination in the anterior margin and in the size and shape of the ventro-marginal row of glands and pores: from left to right: SAIAB 211614, male ML 17 mm; SAIAB 211603, male ML 21 mm; SAIAB 211531, male ML 20 mm; SAIAB 211540, male ML 22 mm. Scale bar 10 mm.

The main diagnostic character of this species is the presence of ventro-marginal subcutaneous glands (Figure 7a) bearing pores (Figure 7b) and contained in fleshy ridges on the anterior part of the ventral mantle—usually one pore per gland, although occasional specimens have one or two anterior glands with two pores, or with the posterior-most gland small and lacking a pore. There are 8–15 (10–13 in 88% of specimens of both sexes) pores per side (Tables A4 and A5), with the number per side usually unequal, and not correlated to ML; pores are dark and highly visible in fresh specimens, but lighter and less obvious after preservation. We confirm Roeleveld's [9] observation that there are no longitudinal grooves linking pores, as evidenced in the original description [16]. Pore and gland function is still unknown. The colour of the ventral surface of the mantle is beige to light brown in preservative (Figure 6), and opalescent pearly white in fresh specimens (see photograph of fresh specimen in [30] (p. 347)), without lateral keels.

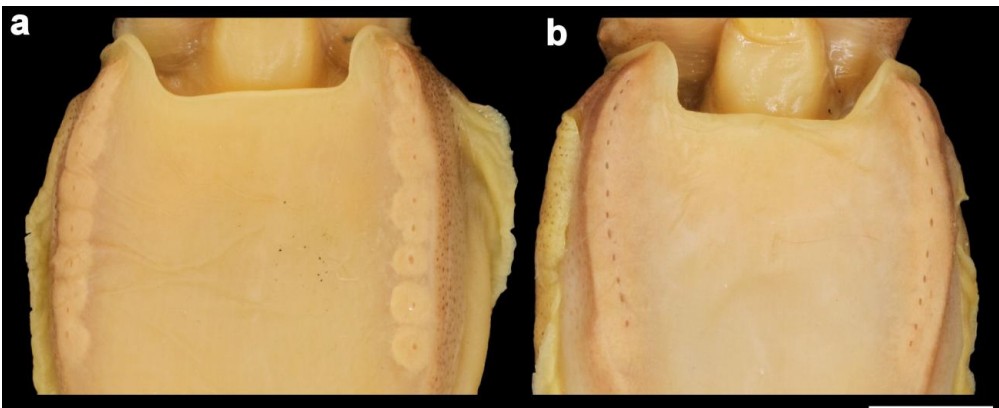

**Figure 7.** Ventro-marginal glands and pores in *Sepia typica*: (**a**) SAIAB 211617, male ML 23 mm, with the skin of ventral mantle removed to expose the glands; (**b**) SAIAB 211607, male ML 22 mm, with sub-cutaneous gland positions indicated by pore openings. Scale bar 10 mm.

The dorsal colour in preservative is light to dark brown or reddish brown (Figures 5 and 8), and mottled greenish-tan with irregular darker reddish-brown patches in fresh specimens (see photograph in [30] (p. 347)). There are usually five main large structures on the dorsal mantle, each structure often a single tubercle but sometimes tubercle clusters or simple turrets of tubercles or papillae. These structures are arranged in a consistent pattern: one large tubercle or papilla (sometimes two very close to each other) anteriorly on the midline; two large structures close to the dorsal midline in the middle of the mantle; and (usually) two smaller structures close to the posterior end of the mantle (Figures 8 and 11). The skin between these structures is usually smooth and shiny (Figure 8), but in some specimens, there are randomly distributed patches of small tubercles (Figures 8 and 11).

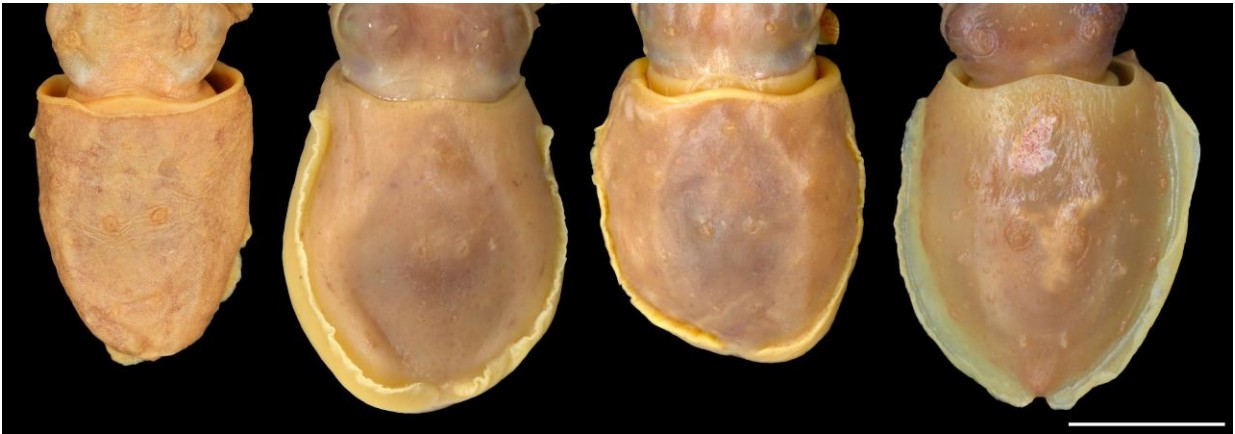

**Figure 8.** *Sepia typica* variation in dorsal mantle structures. From left to right: SAIAB 211627 (male ML 18 mm), mid-dorsal structures large, prominent, anterior and posterior structures weak, skin with small tubercles at random; SAIAB 211541 (male, ML 22 mm), anterior structure very weak, skin smooth; SAIAB 211588 (male ML 20 mm), all structures prominent, skin smooth; SAIAB 211535 (male ML 22 mm), all structures prominent, skin anteriorly with random patches of small tubercles. Scale bar 10 mm.

The head appears elongated because of the membrane joining the arm bases; the width is considerably smaller than the mantle-opening width. The neck and nuchal cartilage are broad; the neck is as wide as the head (Figure 9). In most specimens, there are numerous small tubercles scattered over the dorsal surface of the head (sometimes difficult to see), including a single horizontal row between the eyes and a double row extending from anterior to the eyes down the dorsal surfaces of arms I–III p. (Figures 10 and 11). The eyes are prominent, dorso-lateral, not visibly protruding; there is a prominent supra-orbital tubercle or papillae above the posterior half of the eye (Figures 5, 9 and 10).

The fins are relatively broad (Figure 8), width variable (2.5–18.2%, mean 8.13%, n = 125), ending well before anterior mantle margin (but again rather variable, 5–20%, mean 11.9%, n = 123). Fins separated by a small gap posteriorly (Figure 8).

The tentacular stalk is long (up to 270% ML). The club along the stalk main axis (not perpendicular to it) is relatively short (~17% ML) with small subequal suckers in up to 15 diagonal rows of 2–5 suckers in each (but always four suckers in the middle rows of the club). The protective membranes are small, narrow, and well separated. The natatory membrane is well developed and relatively broad, continuing along tentacular stalk for 0.5–1.0 times the club length (Figure 12).

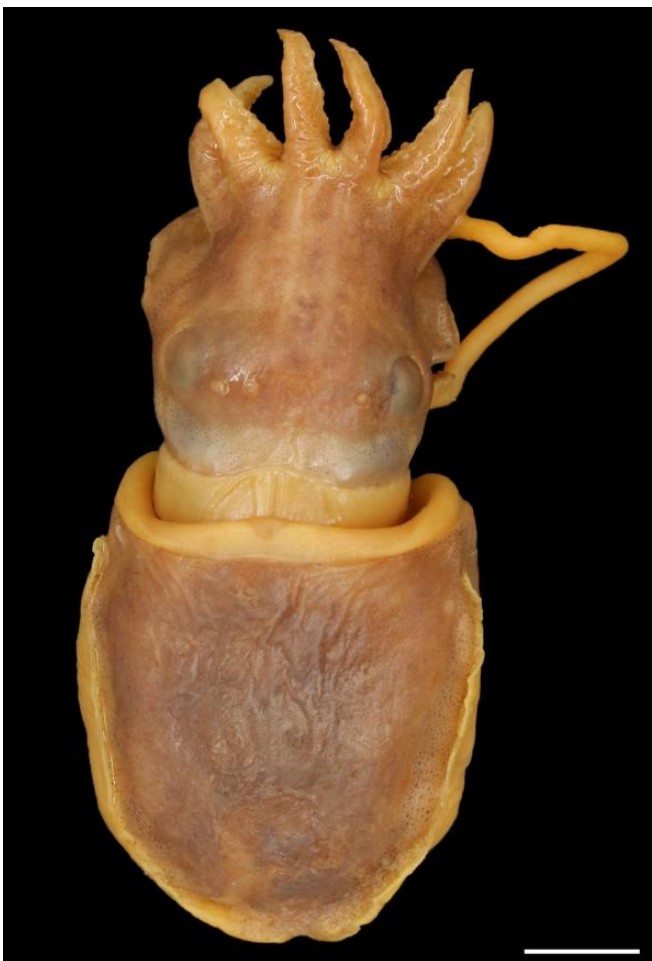

**Figure 9.** *Sepia typica* (SAIAB 211590, female ML 20 mm) showing the nuchal cartilage and the width of the neck relative to the mantle opening. Scale bar 5 mm.

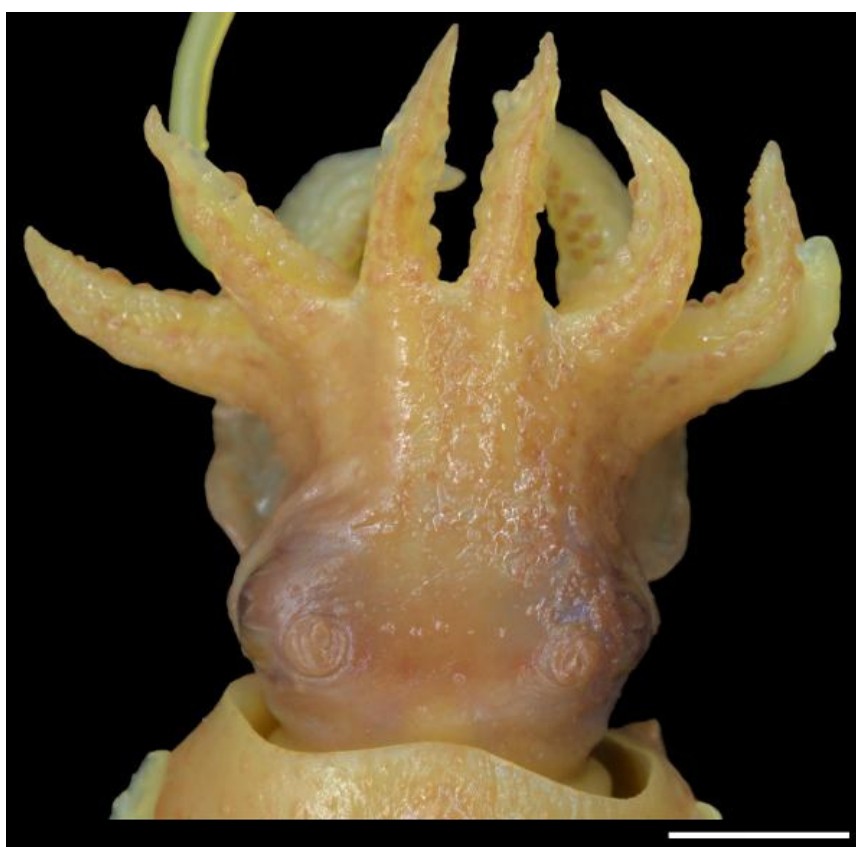

**Figure 10.** Dorsal view of head of. *Sepia typica* (SAIAB 211535 male ML 22 mm) showing prominent supra-orbital papillae and many small tubercles on the head and arms. Scale bar 5 mm.

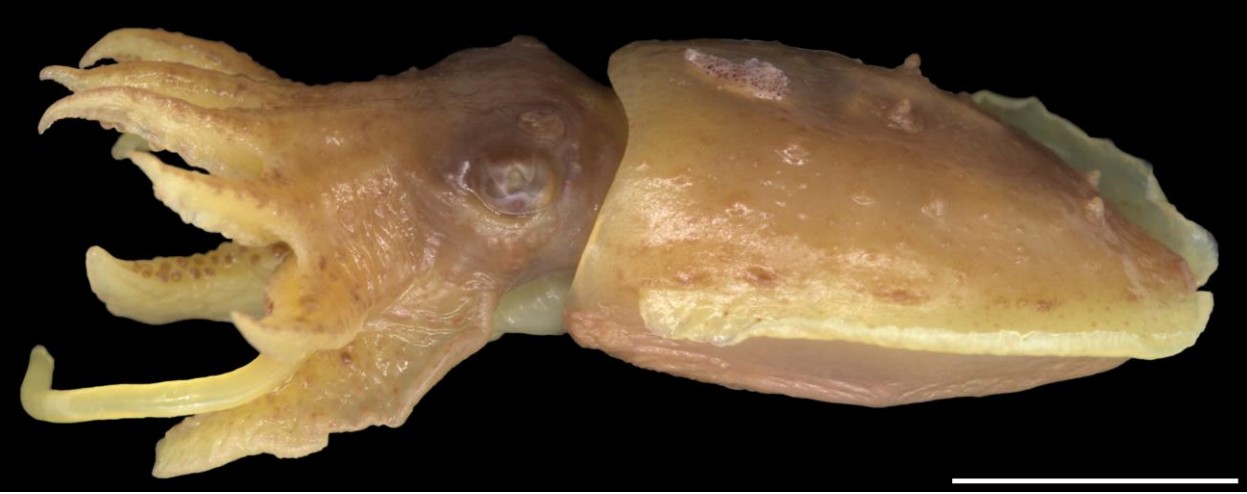

**Figure 11.** Lateral view of *Sepia typica* (SAIAB 211535 male ML 22 mm) showing the prominent structures on the dorsal mantle, supra-orbital papillae and many small tubercles on the head and arms. Scale bar 10 mm.

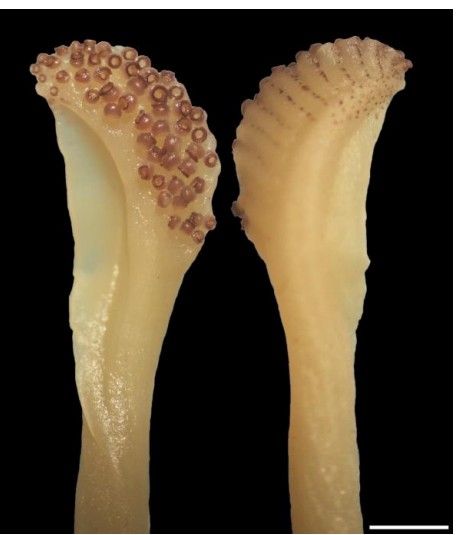

**Figure 12.** Dorsal (**left**) and ventral (**right**) views of left tentacle and club of *Sepia typica* (SAIAB 211554, male ML 21 mm). Scale bar 1 mm.

The arms are robust, stout, and relatively long (Figure 13), with the length varying considerably with age and sex and possibly other factors; there is no single consistent arm formula, but the tendency is for arms to be subequal in females (Table A3), and for IV p. > I p. in males (Table A2). The ventral arms are keeled. There is a strong membrane joining arms for about 40–50% of their length; it is absent between arms IV p. Protective membranes are well developed.

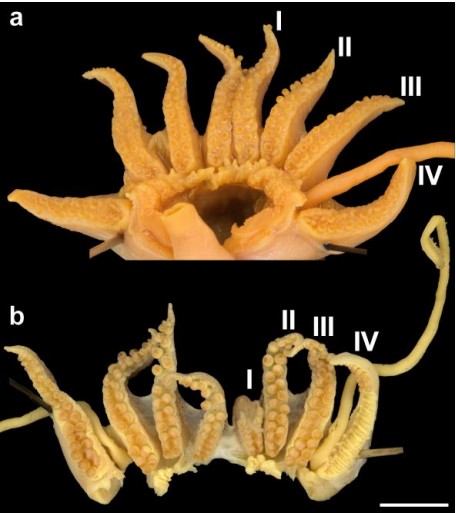

**Figure 13.** Arms of *Sepia typica*: (**a**) Female SAIAB 211684, ML 23 mm; (**b**) Male SAIAB 211655, ML 21 mm. Roman numerals indicate the right member of each arm pair I–IV. Scale bar 5 mm.

Suckers on all arms globose, consistently biserial. Sucker rings with tiny teeth (difficult to see) that are only on the aboral side of arm suckers (Figure 14a), but round the whole circumference of the chitinous ring on club suckers (Figure 14b). Near the tip on all arms of both sexes, sucker diameter decreases abruptly by about 30–50% to a pair of small suckers, with the next pair again noticeably smaller and sucker size gradually decreasing thereafter to minute at the tip (See Figures 15–17). The suckers from (and including) the first abruptly smaller pair to the distal end are here defined as the arm-tip sucker series. The number of suckers in the arm-tip series (ASC1t–ASC4t) is highly variable (3–23; Tables A4 and A5). The arm tips are vulnerable and prone to injury when catching and

subduing prey; therefore, it is likely that this high variability reflects injury and sucker loss rather than genetically-based differences among individuals.

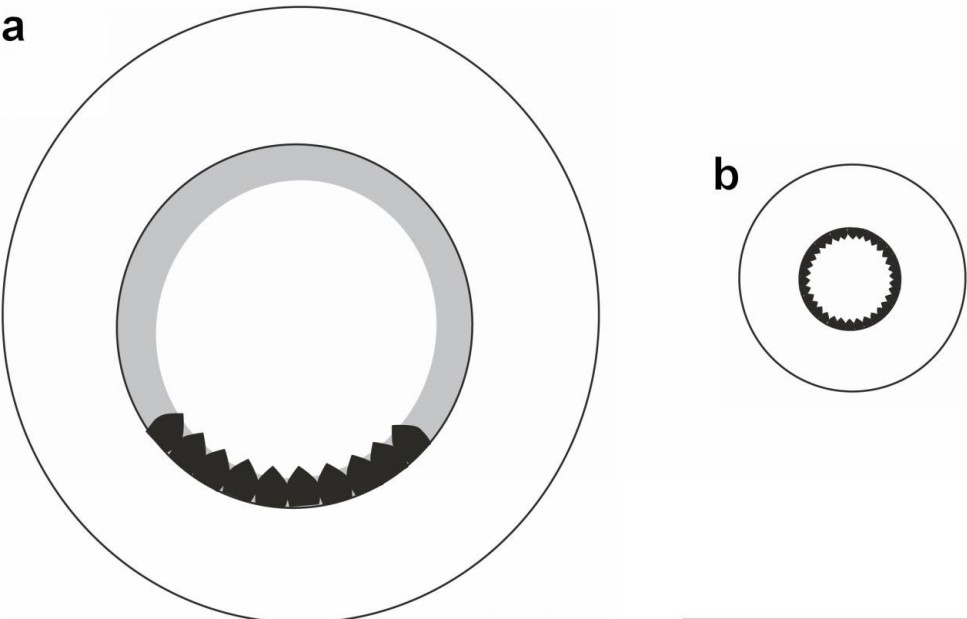

**Figure 14.** Schematic representation of *Sepia typica* arm and club suckers (SAIAB 211654, female ML 21 mm): (**a**) Sucker from third pair on left arm III; (**b**) Sucker from mid club. Scale bar 0.2 mm.

In females the suckers proximal to the arm-tip series subequal in size on all arms. In most males, sucker diameter decreases gradually distally on arms I–III p., but abruptly increases for the last one to four pairs before the arm-tip series, forming a distal cluster of enlarged suckers (Figure 15a). However, in some males, suckers on arms I–III p. are subequal as in females (Figure 15b). Males with subequal (undifferentiated) suckers on arms I–III p. are more common on the East Coast than the West Coast. The unmodified right ventral arm in males (Figure 16), with the basal five to eight sucker pairs large and subequal, the next two to seven sucker pairs are abruptly smaller, followed by a distal cluster of three to six pairs of enlarged suckers before the arm-tip suckers. The hectocotylised left ventral arm (Figure 17) with the basal 40–71% (mean 58.4%) modified, bearing one proximal sucker and eight to 13 pairs of small, marginal suckers (the size decreasing gradually distally), with the ventral series smaller than, and widely separated from, the dorsal series by fleshy transverse folds. The distal portion of the arm is normal, bearing three to seven pairs of enlarged suckers, and two to eleven pairs of small to minute suckers on the arm tip.

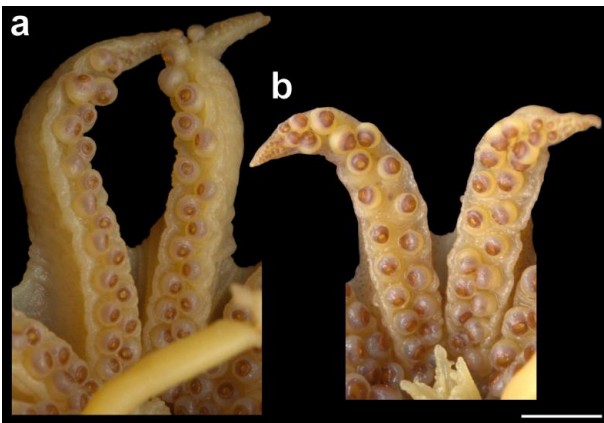

**Figure 15.** *Sepia typica* variation in arms I p. in males: (**a**) SAIAB 211540, ML 22 mm with differentiated arm suckers; (**b**) SAIAB 211557, ML 20 mm with undifferentiated arm suckers. Scale bar 2 mm.

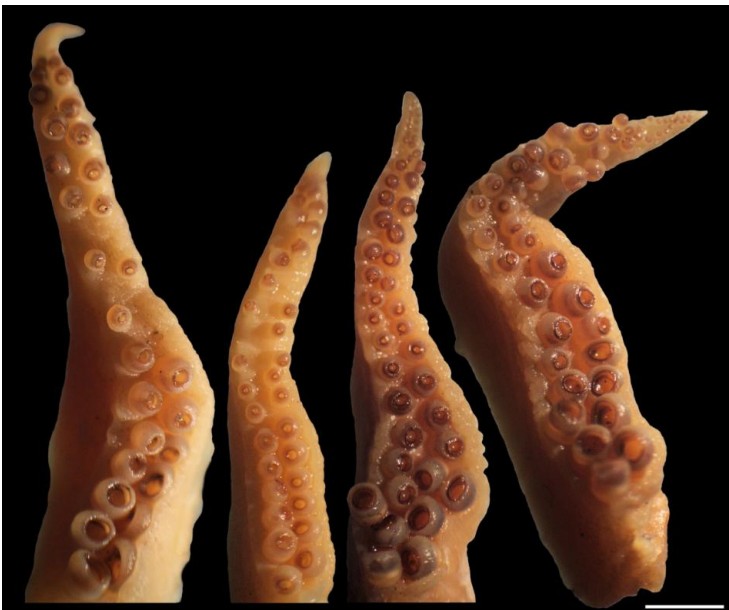

**Figure 16.** *Sepia typica* variation in unmodified right arm IV p of males. From left to right: SAIAB 211659, ML 25 mm, Northwest Area; SAIAB 211640, ML 19 mm, Northwest Area; SAIAB 211586, ML 22 mm, Eastern area; SAIAB 211582, 18 mm, Eastern area. Scale bar 2 mm.

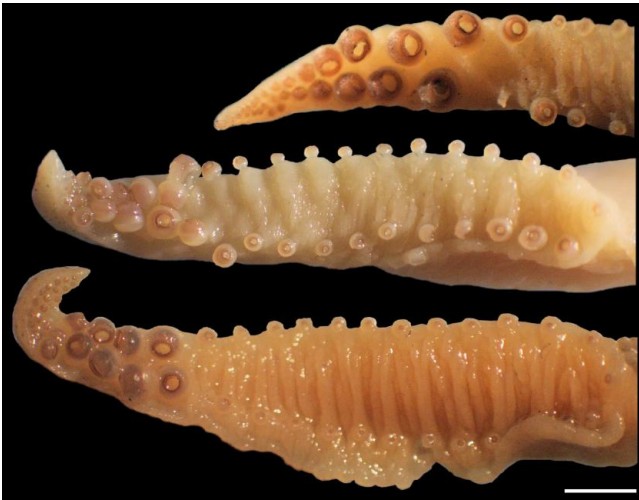

**Figure 17.** *Sepia typica* variation in hectocotylised left arm IV of males. From top to bottom: Arm-tip suckers SAIAB 211659, ML 25 mm, Northwest Area; SAIAB 211640, ML 19 mm, Northwest Area; SAIAB 211586, ML 22 mm, Eastern area. Scale bar 2 mm.

The beaks are small and fragile (Figure 18). The upper beak has a blunt, relatively short, curved and slightly hooked rostrum; its width is roughly equal to length. The jaw edge is straight, and the jaw angle >90°. The hood is high above the crest posteriorly; its edges are strongly curved; the lateral wall posterior edge is slightly curved. The rostrum and hood are dark. Lower beak: rostrum short, blunt, jaw angle rounded without a distinct angle. The hood is low on the crest, which is straight. The posterior edge of the lateral wall is slightly curved and sharp. The rostrum and most of the hood and anterior shoulder region are dark.

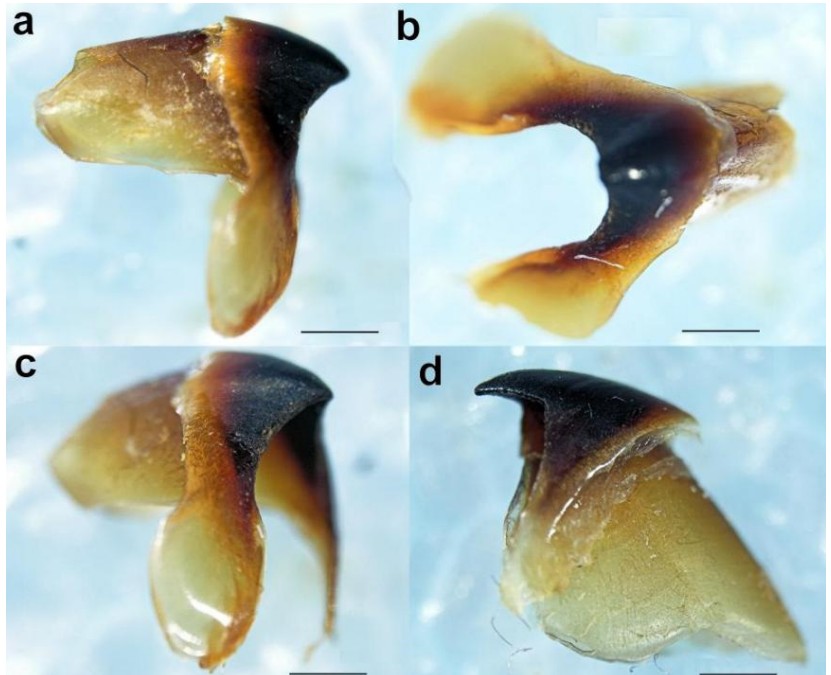

**Figure 18.** Beaks of *Sepia typica*: Lower beak of SAIAB 211629, female ML 18 mm in profile (**a**), top (**b**) and oblique (**c**) aspects; (**d**) Upper beak of SAIAB 211661, male ML 22 mm in profile; Scale bars 1000 μm.

The spermatophores are simple, without any special modifications except a bipartite cement body. The sperm reservoir is quite long (Figure 19a). Radula teeth are simple homodont, but with an unusual tooth shape because the middle tooth and a flanking pair are elongated (Figure 19b). The funnel components of locking cartilages (Figure 20a) are ear-shaped, the internal margin is almost straight, and the central groove is deep and simple without an additional median cleft. The mantle component of locking cartilages is simple and rather small (Figure 20b). The funnel organ is well defined (Figure 21a); the dorsal component has an anterior ridge and papilla, the arms are long and fleshy, and the posterior pads are quite bulky. The ventral part is simple, an elongated oval in shape. The funnel has a well-developed valve (Figure 21b).

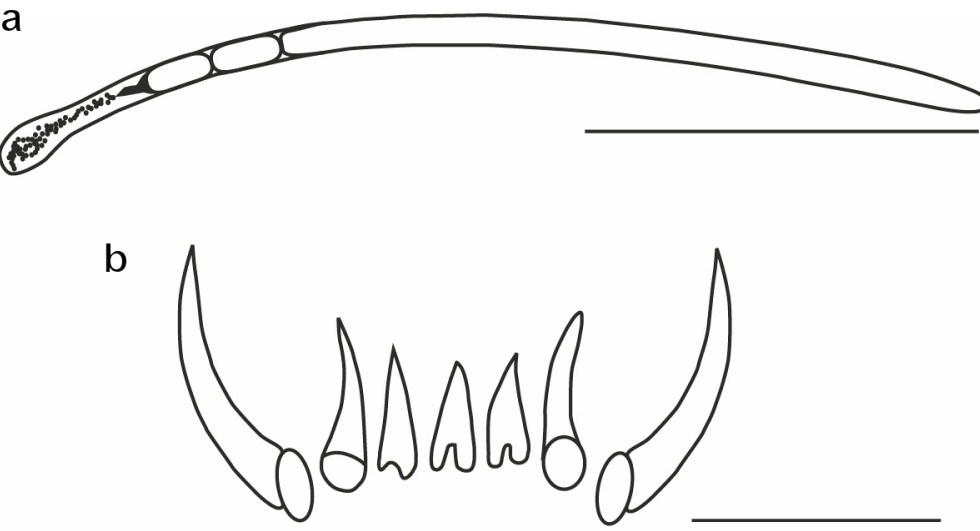

**Figure 19.** *Sepia typica* spermatophore and radula: (**a**) Spermatophore (SAIAB 211627, male ML 18 mm), scale bar 1000 μm; (**b**) Radula (SAIAB 211655, male ML 21 mm), scale bar 500 μm.

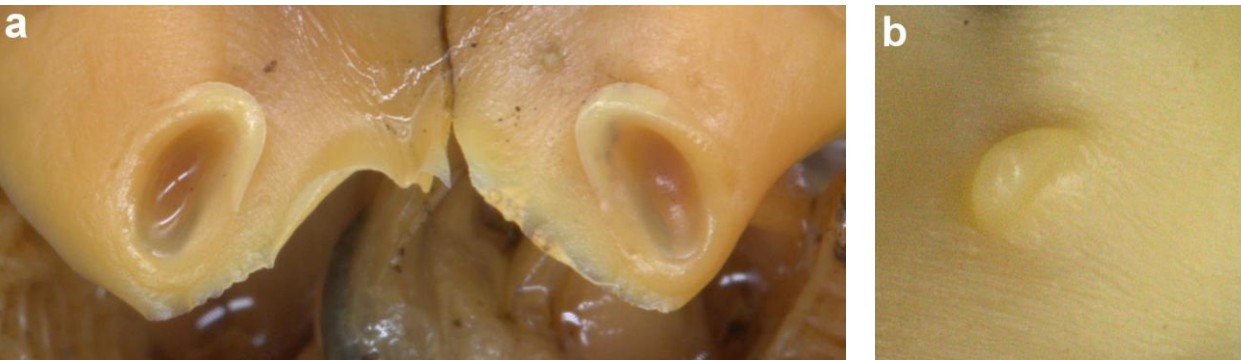

**Figure 20.** *Sepia typica* (SAIAB 211627, male ML 18 mm) funnel/mantle locking mechanism; (**a**) Funnel component; (**b**) Mantle component.

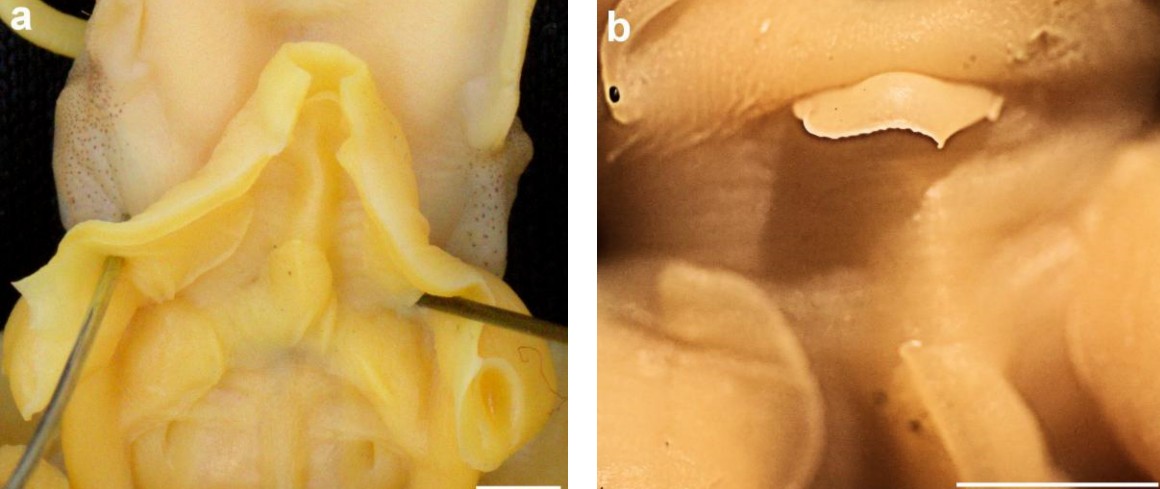

**Figure 21.** *Sepia typica* funnel organ (SAIAB 211656, female ML 19 mm): (**a**) Dorsal component of the funnel organ; (**b**) Enlargement of the funnel valve. Scale bars 2 mm.

The cuttlebone is distinct (Figures 22 and 23); the dorsal shield is chitinous and transparent, without being calcified; the phragmocone (in contrast) is opaque and calcified; the last (anterior) septum is more strongly calcified than the others, borderline concave; the striae are simple and horizontal. The outer cone is broad, transparent, without a spine or mid-dorsal longitudinal ridge. The cuttlebone is sexually dimorphic; in males (Figure 22) the cuttlebone, including the distal part of the phragmocone and inner and outer cone are narrower, with the dorsal shield rounded anteriorly; in females (Figure 23) the dorsal shield, inner and outer cones of the cuttlebone, and anterior part of the phragmocone are generally wider, and are very broad in some individuals (Figure 23b).

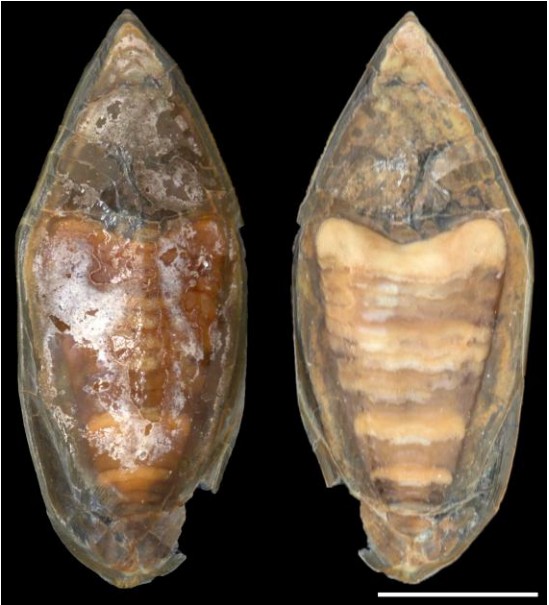

**Figure 22.** Cuttlebone of male *Sepia typica* (SAIAB 211602, ML 19 mm) in dorsal (left) and ventral (right) aspects. Scale bar 10 mm.

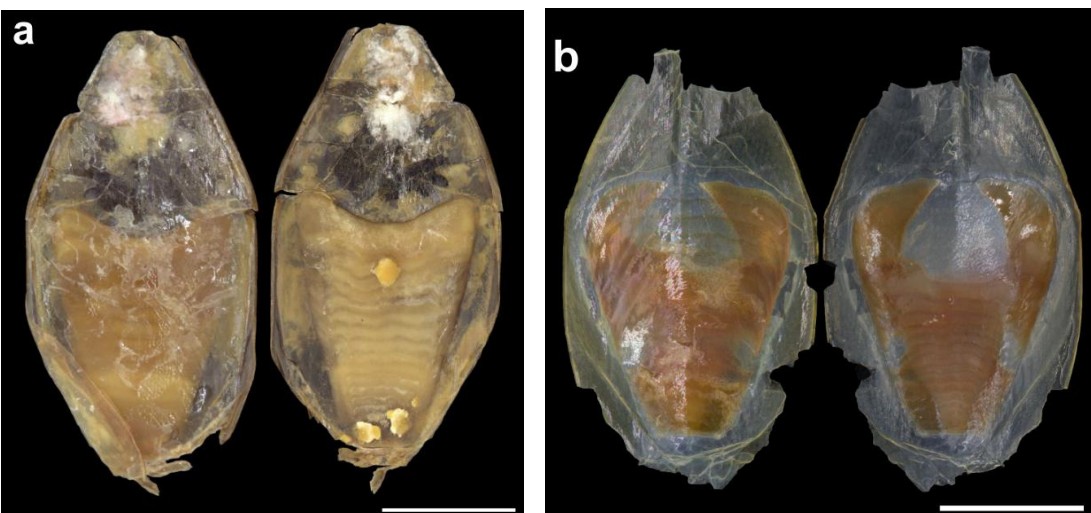

**Figure 23.** Variability in the cuttlebone of female *Sepia typica* in dorsal (left) and ventral (right) aspects: (**a**) SAIAB 211600, ML 19 mm; (**b**) SAIAB 211656, ML 19 mm. Scale bars 5 mm.

**Distribution**: *Sepia typica* is widely distributed (Figure 24), from just north of the Orange River (in southern Namibia) to at least Port Alfred (27° E) on the south-east coast of South Africa, and occurs from shallow waters (4 m [8]) to considerable depths (553 m; this study). It is suspected that the considerable reduction in the calcification of the phragmocone is an adaptation to enable it to live at great depth.

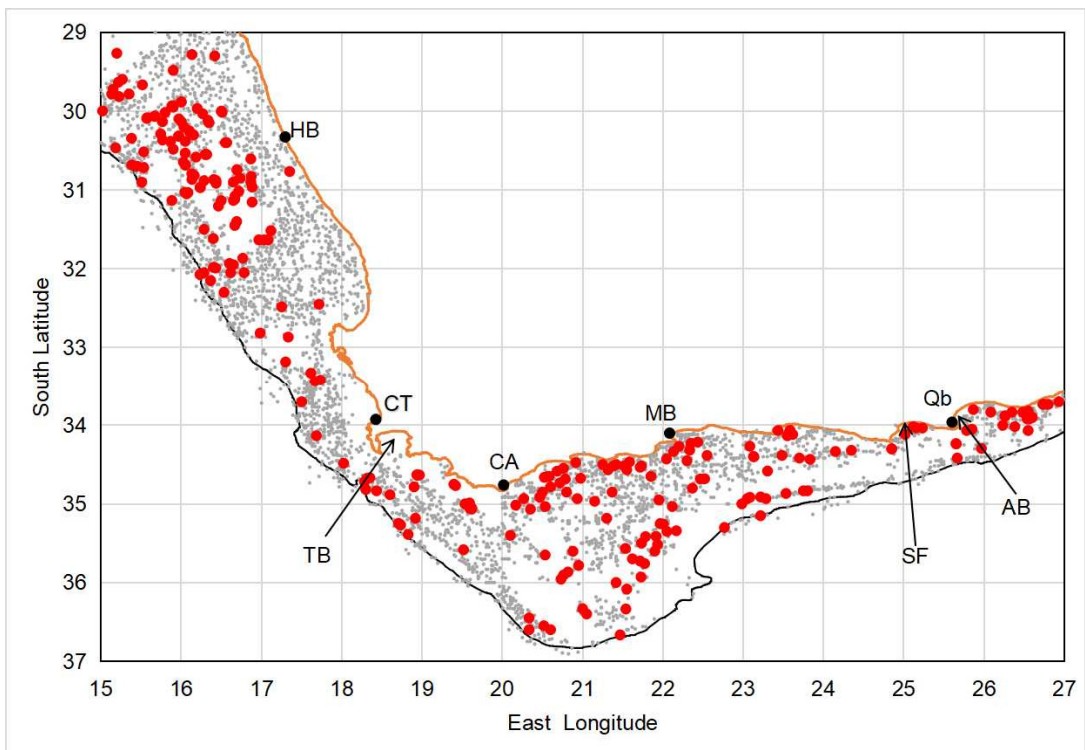

**Figure 24.** Chart showing the positions of all research stations occupied off the west and south coasts of South Africa between 1987 and 2017, and whether *Sepia typica* was recorded (**red dots**) or was not recorded (**small grey dots**). AB = Algoa Bay, CA = Cape Agulhas, CT = Cape Town, HB = Hondeklip Bay, MB = Mossel Bay, Qb = Gqeberha (formally Port Elizabeth), SF = St Francis Bay, TB = Table Bay (type locality). Orange line represents coastline, and black line the 500 m isobath.

## 6. Remarks

*Sepia typica*, an endemic southern African species, was discovered and described almost 150 years ago. Despite this, and the fact that it is a common and easily recognised species, its taxonomic status has been controversial because some important morphological details were lacking. The substantial material available (141 specimens examined) has enabled the present study to confirm, based on morphological characters, that *S. typica* is single well-established species without morphs, variants, or subspecies. All investigated characters are moderately to highly variable (Tables A2–A5).

The perceived differences between west- and east-coast specimens noted in the literature [5–8] (and summarised in Section 1, Introduction) stems from the limited east-coast (♂2, ♀1) material that was available to those authors. Roeleveld [8] reviewed all of the *S. typica* material that was known at the time, namely her collection of 27 males and 44 females and the individuals described in the literature [5–7,16,21,22]. She concluded that Chun's specimen (at the time the only intact male known from the East Coast) differed from west-coast specimens only in having fewer suckers (13 pairs = 26 suckers) on the right arm IV, of which none were enlarged distally.

Sucker numbers on the unmodified right arm IV of the males included in the present study (Table A4) are as follows: Northwest Area: 40–54 (mean 45.7 ± 4.2, n = 20); Central Area: 35–55 (mean 44.7 ± 6.0, n = 22); Eastern Area: 37–52 (mean 45.4 ± 3.9, n = 21). It is therefore clear that this particular character cannot be used for any systematic decisions concerning the species. Figure 16 shows that not all males have obviously enlarged distal suckers on the right arm IV; thus, the only difference remaining between Chun's specimen is the very low sucker count for that arm, which is substantially lower than for any of the males in this study collected from the same locality as his specimen. Thus, we conclude that Chun's specimen may have been damaged, or is aberrant in this character and does not represent an "eastern form" of *S. typica*.

Other characters investigated in the present study (e.g., dorsal and ventral margins of the mantle) are also variable in other species of small sepiids [2–4].

Earlier studies [3,4] noted changes of cuttlebones of small sepiids during preservation and storage, and interpretation of those changes from the moment of capture to the moment of analysis and illustration. The present study contributes substantially to these findings. Steenstrup's original illustration of the cuttlebone of *Sepia typica* ([16], Pl. 1 Figures 3–5), our illustration of the same specimen (Figure A2) and similar illustrations by Roeleveld (Figure 17c,d of [8]) suggest that the phragmacone of the illustrated cuttlebones was considerably damaged (Figures 22 and 23). This damage may lead to incorrect conclusions being drawn about reduction in the phragmocone, and may influence phylogenetic considerations (important in some publications, e.g., [31,32]). It should also be noted that the colour of the background can affect the impression of the structure of the cuttlebone; compare for example the photographs of the first published image of the complete cuttlebone of *S. typica* (Figures 36 and 37 in Ref [4]), with Figure 22 in this paper taken of the same specimen. Figure 22 was taken on a black background, which is standard for systematic work and required by many journals, and gives the impression of a substantial cuttlebone. In contrast, Lipinski's [4] photograph of the same specimen, taken on a pale blue background, clearly illustrates the insubstantial, transparent nature of the outer cone.

Results of the present study serve as purely morphology-based suggestion of *Sepia typica* identity and phylogenetic links. Ultimately, to fully resolve the status of *S. typica* and reach more advanced conclusions requires molecular analyses of a large number of specimens.

**Supplementary Materials:** The following supporting information can be downloaded at: https://zenodo.org/record/7414835#.Y5Hojn1BxPY, Excel spreadsheet containing the meristic counts and morphometric measurements for each specimen examined.

**Author Contributions:** A.J.R. and R.W.L. undertook the statistical analyses, R.W.L. and M.R.L. contributed equally to all other aspects of the manuscript. All authors have read and agreed to the published version of the manuscript.

**Funding:** This research received no external funding.

**Institutional Review Board Statement:** Not applicable (animal museum specimens used).

**Informed Consent Statement:** Not applicable.

**Data Availability Statement:** Meristic counts and morphometric measurements for each specimen examined are available as Supplementary Material.

**Acknowledgments:** We would like to thank our sea-going colleagues (notably Shaz du Plessis and Oddgeir Alvheim) for collecting some of the material used in this study. We would like to thank Tom Schiøtte (Zoological Museum, Copenhagen, Denmark) for help with the type specimen of *Sepia typica* (photographs) and Marcin Białowąs (Sea Fisheries, Gdynia, Poland) for access to the Nikon SMZ18 stereomicroscope. We also recognize the substantial input of the referees, which improved the manuscript. Finally, we thank most sincerely our colleagues from the Institute of Marine Research (Bergen, Norway), FAO (Rome, Italy), and Department of Forestry, Fisheries and Environment (DFFE; Cape Town, South Africa).

**Conflicts of Interest:** The authors declare no conflict of interest.

## Appendix A

**Table A1.** Measurements and counts used in this study. An asterisk (*) denotes measurements or counts defined here for the first time, other definitions follow Refs [2–4]. See Description for details of the various sucker fields on the arms. TW in gm. AMH, FIa, P1 and all sucker diameters to nearest 0.1 mm. FW to the nearest 0.5 mm. All other measurements to the nearest 1.0 mm.

| Abbreviation | Description or Definition |
|---|---|
| AL1–AL4 | **Arm Length**: length of the right (rt) or left (lt) of each designated (1 to 4) arm pair. |
| AMH | **Anterior Mantle to Head**: length of anterior projection of dorsal mantle margin. |
| AS1–AS4 | **Arm Sucker Diameter**: replaced by AS1c–AS4c; AS1pl–AS4pl and AS1ps–AS4ps (males) or AS1pl–AS4pl (females). |
| AS1c–AS4c | * **Arm Sucker Diameter (cluster)**: diameter of the largest sucker in the distal cluster on the right (rt) or left (lt) arm of each designated (1 to 4) arm pair (males only). |
| AS1pl–AS4pl | * **Arm Sucker Diameter (proximal large)**: diameter of the largest sucker in the proximal sucker field on the right (rt) or left (lt) arm of each designated (1 to 4) arm pair. |
| AS1ps–AS4ps | * **Arm Sucker Diameter (proximal small)**: diameter of the smallest sucker in the proximal sucker field on the right (rt) or left (lt) arm of each designated (1 to 4) arm pair (males only). |
| ASC1–ASC4 | **Arm Sucker Count**: total number of suckers on the right (rt) or left (lt) arm of each designated (1 to 4) arm pair; sum of the counts of the number of suckers in each defined sucker field. |
| ASC1c–ASC4c | * **Arm Sucker Count (cluster)**: count of suckers in the distal cluster on the right (rt) or left (lt) arm of each designated (1 to 4) arm pair (males only). |
| ASC1nt–ASC4nt | * **Arm Sucker Count (no tip)**: total number of suckers on the right (rt) or left (lt) arm of each designated (1 to 4) arm pair excluding the suckers at the distal tip. |
| ASC1p–ASC4p | * **Arm Sucker Count (proximal)**: count of the suckers in the proximal sucker field on the right (rt) or left (lt) arm of each designated (1 to 4) arm pair (males only). |
| ASC1t–ASC4t | * **Arm Sucker Count (tip)**: count of the suckers at the distal tip of the right (rt) or left (lt) arm of each designated (1 to 4) arm pair. |
| ASC4m | * **Arm Sucker Count 4 (mid)**: count of the suckers in the mid-arm sucker field on the right Arm IV (males only). |
| ASl4 | **Arm Sucker left 4**: diameter of largest sucker on the hectocotylised left ventral arm (males only). |
| ASl4m | **Arm Sucker left 4 minimum**: diameter of smallest sucker on the hectocotylised left ventral arm (males only). |

Table A1. *Cont.*

| Abbreviation | Description or Definition |
|---|---|
| ClRC | **Club Row Count**: number of suckers per transverse row across the middle of a tentacular club. |
| ClS | **Club Sucker Diameter**: diameter of the largest sucker on a tentacular club. |
| CS# | **Club Sucker Count**: total number of suckers on a tentacular club. |
| FFu | **Free Funnel Length**: length of the funnel along its dorsal midline. |
| FL | **Fin Length**: length of the base of the left fin along the curve of mantle. |
| FIa | **Fin Insertion Anterior**: distance from the anterior mantle margin to the anterior junction of the fin and mantle. |
| FIp | **Fin Insertion Posterior**: distance between the junction of the left and right fins with the mantle. |
| FuL | **Funnel Length**: length of the funnel along its ventral midline. |
| FW | **Fin Width**: width of a single fin at its widest point. |
| HcL | **Hectocotylus Length**: length of the hectocotylised (left ventral) arm (males only). |
| Hect | **Hectocotylus sucker count**: total number of suckers on hectocotylised (left ventral) arm; sum of HSCm + HSCc + HSCt (males only). |
| HL | **Head Length**: length of the head from the nuchal cartilage to the interbranchial membrane |
| HSCm | * **Hectocotylus Sucker Count (modified)**: count of suckers on modified part of hectocotylised (left ventral) arm (males only). |
| HSCt | * **Hectocotylus Sucker Count (tip)**: count of suckers in the tip field of the hectocotylised (left ventral) arm (males only). |
| HSCu | * **Hectocotylus Sucker Count (unmodified)**: count of the unmodified suckers on the hectocotylised (left ventral) arm distal to the modified portion and proximal to the suckers of the tip field (males only). |
| HW | **Head Width**: the greatest width of the head. |
| L | **Length of Cuttlebone**: measured along midline. |
| MHL | **Modified Hectocotylus Length**: of the modified (proximal) portion of the hectocotylus. |
| ML | **Mantle Length**: length of mantle along the dorsal midline. |
| MLv | **Mantle Length (ventral)**: length of mantle along the ventral midline. |
| P#-lt and P#-rt | * **Pore Count**: count of the pore openings along the left (lt) or right (rt) antero-marginal ridge. |
| PD-lt and PD-rt | * **Pore Distance**: measured from the mantle edge to the anterior edge of the first pore opening in the left (lt) or right (rt) antero-marginal ridge. |
| PR-lt and PR-rt | * **Pore Row Length**: measured in a direct line from the anterior edge of the first (anterior) pore opening to the posterior edge of the last (posterior) pore opening in the left (lt) or right (rt) antero-marginal ridge. |
| Tcl | **Tentacular Club Length**: measured from the basal sucker to the tip of club. |
| TL | **Tentacle Length**: measured from point of emergence to from tentacular sac to the tip of the club. |
| TrRC | **Transverse Row Count**: number of transverse sucker rows along the longitudinal length of the club. |
| TW | **Total Weight**: measured in grams. |

Table A2. Morphometric measurements of male *Sepia typica*. Sample size (*n*), minimum (Min), maximum (Max), arithmetic mean (Mean) and standard deviation of the mean (Std) are presented by region and for all regions combined for each character (See Table A1 for character abbreviations). TW (g) and ML (mm), other characters given as percentage of ML, except for MHL, which is given as percentage of HcL.

| Area | Northwest Area | | | | | Central Area | | | | | Eastern Area | | | | | Areas Combined | | | | |
|---|---|---|---|---|---|---|---|---|---|---|---|---|---|---|---|---|---|---|---|---|
| Char-acter | n | Min | Max | Mean | Std | n | Min | Max | Mean | Std | n | Min | Max | Mean | Std | n | Min | Max | Mean | Std |
| TW | 21 | 0.91 | 4.45 | 2.66 | 0.79 | 23 | 1.40 | 3.46 | 2.27 | 0.54 | 23 | 0.57 | 3.65 | 2.32 | 0.83 | 67 | 0.57 | 4.45 | 2.41 | 0.74 |
| ML | 22 | 13 | 25 | 19.7 | 2.8 | 23 | 17 | 23 | 19.1 | 2.1 | 23 | 12 | 22 | 18.1 | 2.6 | 68 | 12 | 25 | 19.0 | 2.6 |
| MLv | 22 | 78 | 106 | 89.7 | 6.9 | 22 | 82 | 100 | 90.0 | 4.6 | 23 | 81 | 111 | 89.9 | 6.6 | 67 | 78 | 111 | 89.9 | 6.0 |
| AL1 | 21 | 43 | 62 | 51.5 | 5.0 | 22 | 35 | 53 | 45.7 | 4.4 | 21 | 40 | 60 | 50.2 | 5.9 | 64 | 35 | 62 | 49.1 | 5.6 |
| AL2 | 22 | 44 | 61 | 53.0 | 4.9 | 22 | 40 | 61 | 49.5 | 4.9 | 21 | 44 | 65 | 52.9 | 5.0 | 65 | 40 | 65 | 51.7 | 5.1 |

**Table A2.** *Cont.*

| Area | Northwest Area | | | | | Central Area | | | | | Eastern Area | | | | | Areas Combined | | | | |
|---|---|---|---|---|---|---|---|---|---|---|---|---|---|---|---|---|---|---|---|---|
| Char-acter | n | Min | Max | Mean | Std | n | Min | Max | Mean | Std | n | Min | Max | Mean | Std | n | Min | Max | Mean | Std |
| AL3 | 22 | 47 | 67 | 57.9 | 5.2 | 22 | 48 | 61 | 55.5 | 3.9 | 21 | 50 | 69 | 57.0 | 4.7 | 65 | 47 | 69 | 56.8 | 4.6 |
| AL4 | 22 | 38 | 80 | 59.7 | 9.7 | 22 | 52 | 79 | 62.7 | 7.0 | 21 | 52 | 89 | 66.4 | 9.7 | 65 | 38 | 89 | 62.9 | 9.2 |
| AMH | 17 | 0.0 | 7.3 | 3.35 | 2.66 | 23 | 0.0 | 7.8 | 3.65 | 1.96 | 21 | 1.5 | 7.6 | 3.99 | 1.59 | 61 | 0.0 | 7.8 | 3.68 | 2.05 |
| AS1c | 20 | 3.0 | 4.6 | 3.86 | 0.41 | 22 | 3.3 | 5.0 | 3.95 | 0.45 | 21 | 2.9 | 4.4 | 3.78 | 0.39 | 63 | 2.9 | 5.0 | 3.86 | 0.42 |
| AS1pl | 21 | 2.7 | 3.9 | 3.45 | 0.34 | 22 | 2.5 | 3.9 | 3.34 | 0.37 | 22 | 2.5 | 4.4 | 3.45 | 0.46 | 65 | 2.5 | 4.4 | 3.41 | 0.39 |
| AS1ps | 19 | 1.7 | 3.9 | 2.87 | 0.62 | 22 | 2.2 | 3.7 | 3.01 | 0.45 | 21 | 2.4 | 3.9 | 3.10 | 0.38 | 62 | 1.7 | 3.9 | 3.00 | 0.49 |
| AS2c | 21 | 3.1 | 5.4 | 4.55 | 0.52 | 22 | 3.5 | 5.5 | 4.42 | 0.52 | 22 | 2.9 | 5.4 | 4.50 | 0.57 | 65 | 2.9 | 5.5 | 4.49 | 0.53 |
| AS2pl | 21 | 3.1 | 4.4 | 3.66 | 0.37 | 22 | 3.0 | 4.4 | 3.74 | 0.40 | 22 | 3.3 | 4.7 | 3.93 | 0.37 | 65 | 3.0 | 4.7 | 3.78 | 0.39 |
| AS2ps | 21 | 2.2 | 3.9 | 3.24 | 0.47 | 22 | 2.4 | 4.1 | 3.35 | 0.46 | 22 | 2.5 | 4.1 | 3.45 | 0.40 | 65 | 2.2 | 4.1 | 3.35 | 0.45 |
| AS3c | 22 | 3.1 | 4.7 | 3.96 | 0.41 | 22 | 3.5 | 5.0 | 4.02 | 0.41 | 22 | 2.9 | 4.4 | 3.91 | 0.36 | 66 | 2.9 | 5.0 | 3.96 | 0.39 |
| AS3pl | 22 | 3.1 | 4.7 | 3.99 | 0.42 | 22 | 3.5 | 5.6 | 4.17 | 0.53 | 22 | 3.5 | 5.6 | 4.32 | 0.51 | 66 | 3.1 | 5.6 | 4.16 | 0.50 |
| AS3ps | 22 | 1.7 | 3.9 | 3.07 | 0.48 | 22 | 2.2 | 4.1 | 3.26 | 0.50 | 22 | 2.4 | 4.4 | 3.35 | 0.44 | 66 | 1.7 | 4.4 | 3.22 | 0.48 |
| AS4c | 19 | 1.9 | 3.5 | 2.94 | 0.37 | 22 | 2.4 | 3.5 | 2.97 | 0.36 | 20 | 1.8 | 3.8 | 2.94 | 0.50 | 61 | 1.8 | 3.8 | 2.95 | 0.41 |
| AS4pl | 20 | 3.1 | 5.9 | 4.75 | 0.73 | 22 | 4.0 | 6.3 | 4.92 | 0.65 | 22 | 4.0 | 6.2 | 5.07 | 0.54 | 64 | 3.1 | 6.3 | 4.92 | 0.64 |
| AS4ps | 19 | 1.4 | 2.8 | 2.23 | 0.35 | 22 | 1.1 | 3.0 | 2.27 | 0.57 | 20 | 1.5 | 2.8 | 2.30 | 0.31 | 61 | 1.1 | 3.0 | 2.26 | 0.42 |
| ASl4 | 21 | 1.9 | 3.6 | 2.84 | 0.37 | 22 | 2.0 | 3.5 | 2.93 | 0.39 | 22 | 2.0 | 3.7 | 2.94 | 0.39 | 65 | 1.9 | 3.7 | 2.90 | 0.38 |
| ASl4m | 21 | 0.8 | 1.5 | 1.10 | 0.18 | 22 | 0.9 | 1.2 | 1.06 | 0.11 | 22 | 0.5 | 1.5 | 1.05 | 0.25 | 65 | 0.5 | 1.5 | 1.07 | 0.19 |
| ClS | 14 | 0.9 | 1.4 | 1.14 | 0.18 | 22 | 0.9 | 1.7 | 1.20 | 0.22 | 19 | 0.9 | 1.6 | 1.13 | 0.18 | 55 | 0.9 | 1.7 | 1.16 | 0.19 |
| FFu | 22 | 14 | 29 | 22.8 | 3.9 | 23 | 11 | 29 | 17.1 | 4.2 | 22 | 14 | 33 | 21.5 | 4.1 | 67 | 11 | 33 | 20.4 | 4.7 |
| FIa | 22 | 7.0 | 18.0 | 12.64 | 3.61 | 23 | 6.0 | 16.0 | 9.70 | 2.67 | 21 | 5.0 | 17.0 | 11.57 | 2.56 | 66 | 5.0 | 18.0 | 11.27 | 3.19 |
| FIp | 21 | 0.0 | 6.7 | 2.69 | 1.78 | 23 | 0.5 | 8.3 | 3.70 | 1.91 | 20 | 1.6 | 7.5 | 3.77 | 1.28 | 64 | 0.0 | 8.3 | 3.39 | 1.74 |
| FL | 22 | 78 | 115 | 101.7 | 10.0 | 23 | 81 | 124 | 108.6 | 9.5 | 21 | 95 | 126 | 107.4 | 9.2 | 66 | 78 | 126 | 105.9 | 9.9 |
| FuL | 22 | 36 | 63 | 50.8 | 6.7 | 23 | 39 | 59 | 46.2 | 5.1 | 22 | 42 | 56 | 46.4 | 3.5 | 67 | 36 | 63 | 47.8 | 5.6 |
| FW | 22 | 3.8 | 18.2 | 7.82 | 3.18 | 23 | 4.8 | 15.2 | 9.14 | 2.96 | 21 | 4.5 | 11.8 | 7.68 | 1.75 | 66 | 3.8 | 18.2 | 8.23 | 2.76 |
| HcL | 22 | 45 | 75 | 58.7 | 7.9 | 22 | 50 | 61 | 56.8 | 2.6 | 21 | 53 | 74 | 61.3 | 6.2 | 65 | 45 | 75 | 58.9 | 6.2 |
| HL | 22 | 56 | 77 | 65.4 | 6.2 | 23 | 55 | 71 | 63.0 | 4.8 | 23 | 55 | 73 | 64.5 | 5.4 | 68 | 55 | 77 | 64.3 | 5.5 |
| HW | 22 | 48 | 69 | 57.1 | 5.3 | 23 | 45 | 61 | 53.6 | 4.3 | 23 | 47 | 67 | 54.8 | 4.6 | 68 | 45 | 69 | 55.1 | 4.9 |
| MHL | 22 | 47 | 71 | 58.5 | 6.9 | 22 | 40 | 64 | 57.3 | 6.1 | 21 | 43 | 70 | 58.0 | 6.7 | 65 | 40 | 71 | 58.4 | 6.5 |
| PD-lt | 22 | 1.8 | 13.9 | 8.07 | 3.68 | 21 | 2.9 | 11.8 | 7.11 | 2.34 | 20 | 1.0 | 12.1 | 7.67 | 2.70 | 63 | 1.0 | 13.9 | 7.62 | 2.96 |
| PD-rt | 22 | 2.7 | 14.7 | 9.62 | 3.15 | 23 | 3.6 | 12.4 | 7.96 | 2.51 | 21 | 1.5 | 14.7 | 8.12 | 3.27 | 66 | 1.5 | 14.7 | 8.57 | 3.03 |
| PR-lt | 22 | 31 | 58 | 43.2 | 7.6 | 21 | 33 | 54 | 41.1 | 5.2 | 19 | 36 | 51 | 42.9 | 4.1 | 62 | 31 | 58 | 42.4 | 5.9 |
| PR-rt | 22 | 33 | 56 | 43.68 | 6.49 | 23 | 28 | 50 | 42.78 | 5.09 | 21 | 29 | 54 | 42.05 | 5.56 | 66 | 28 | 56 | 42.85 | 5.69 |
| Tcl | 14 | 13 | 21 | 16.0 | 2.5 | 22 | 11 | 24 | 15.7 | 2.9 | 19 | 10 | 23 | 16.1 | 2.7 | 55 | 10 | 24 | 15.9 | 2.7 |
| TL | 14 | 150 | 217 | 186.4 | 21.7 | 21 | 65 | 229 | 133.6 | 42.3 | 19 | 67 | 206 | 166.2 | 32.8 | 54 | 65 | 229 | 158.7 | 40.4 |

**Table A3.** Morphometric measurements of female *Sepia typica*. Sample size (*n*), minimum (Min), maximum (Max), arithmetic mean (Mean) and standard deviation of the mean (Std) are presented by region and for all regions combined for each character (See Table A1 for character abbreviations). TW (g) and ML (mm), other characters given as percentage of ML.

| Area | Northwest Area | | | | | Central Area | | | | | Eastern Area | | | | | Areas Combined | | | | |
|---|---|---|---|---|---|---|---|---|---|---|---|---|---|---|---|---|---|---|---|---|
| Char-acter | n | Min | Max | Mean | Std | n | Min | Max | Mean | Std | n | Min | Max | Mean | Std | n | Min | Max | Mean | Std |
| TW | 18 | 0.78 | 5.05 | 3.20 | 1.15 | 15 | 0.90 | 4.07 | 2.30 | 0.72 | 26 | 1.26 | 4.48 | 2.90 | 0.88 | 59 | 0.78 | 5.05 | 2.84 | 0.98 |
| ML | 18 | 12 | 25 | 19.9 | 3.3 | 15 | 12 | 23 | 18.0 | 2.8 | 26 | 14 | 24 | 19.2 | 2.3 | 59 | 12 | 25 | 19.1 | 2.8 |
| MLv | 18 | 83 | 100 | 91.6 | 4.8 | 13 | 75 | 106 | 90.6 | 8.6 | 26 | 75 | 100 | 90.3 | 6.0 | 57 | 75 | 106 | 90.8 | 6.3 |
| AL1 | 18 | 43 | 61 | 49.4 | 4.3 | 14 | 36 | 61 | 45.6 | 7.5 | 26 | 35 | 58 | 44.1 | 5.0 | 58 | 35 | 61 | 46.1 | 5.9 |
| AL2 | 18 | 40 | 58 | 49.2 | 4.3 | 15 | 36 | 53 | 44.1 | 4.7 | 26 | 40 | 53 | 44.0 | 3.4 | 59 | 36 | 58 | 45.6 | 4.6 |
| AL3 | 18 | 43 | 61 | 50.0 | 5.0 | 15 | 41 | 53 | 45.9 | 4.2 | 26 | 38 | 50 | 44.4 | 3.6 | 59 | 38 | 61 | 46.5 | 4.8 |
| AL4 | 18 | 33 | 69 | 54.8 | 8.0 | 15 | 42 | 67 | 53.4 | 7.9 | 26 | 40 | 68 | 52.7 | 6.8 | 59 | 33 | 69 | 53.5 | 7.4 |

**Table A3.** *Cont.*

| Area | Northwest Area | | | | | Central Area | | | | | Eastern Area | | | | | Areas Combined | | | | |
|---|---|---|---|---|---|---|---|---|---|---|---|---|---|---|---|---|---|---|---|---|
| Char-acter | n | Min | Max | Mean | Std | n | Min | Max | Mean | Std | n | Min | Max | Mean | Std | n | Min | Max | Mean | Std |
| AMH | 13 | 0.0 | 6.7 | 4.19 | 1.90 | 11 | 0.0 | 8.9 | 4.34 | 2.17 | 24 | 1.6 | 7.5 | 3.96 | 1.55 | 48 | 0.0 | 8.9 | 4.11 | 1.76 |
| AS1pl | 18 | 2.4 | 4.2 | 2.90 | 0.44 | 15 | 2.6 | 4.7 | 3.27 | 0.58 | 26 | 2.2 | 3.6 | 2.90 | 0.35 | 59 | 2.2 | 4.7 | 2.99 | 0.47 |
| AS2pl | 18 | 2.4 | 4.2 | 3.01 | 0.45 | 15 | 2.6 | 3.5 | 3.05 | 0.35 | 26 | 2.5 | 3.9 | 3.05 | 0.36 | 59 | 2.4 | 4.2 | 3.04 | 0.38 |
| AS3pl | 18 | 2.6 | 5.0 | 3.07 | 0.52 | 15 | 2.6 | 4.0 | 3.24 | 0.46 | 26 | 2.5 | 3.9 | 2.90 | 0.41 | 59 | 2.5 | 5.0 | 3.04 | 0.47 |
| AS4pl | 18 | 2.6 | 4.2 | 3.08 | 0.37 | 15 | 2.6 | 3.5 | 3.05 | 0.35 | 26 | 2.4 | 3.7 | 2.91 | 0.30 | 59 | 2.4 | 4.2 | 3.00 | 0.34 |
| ClS | 16 | 0.9 | 1.7 | 1.23 | 0.25 | 12 | 1.1 | 2.0 | 1.34 | 0.29 | 20 | 0.8 | 1.7 | 1.22 | 0.24 | 48 | 0.8 | 2.0 | 1.25 | 0.26 |
| FFu | 18 | 13 | 31 | 21.7 | 4.6 | 15 | 13 | 30 | 21.1 | 5.4 | 25 | 13 | 29 | 20.6 | 3.4 | 58 | 13 | 31 | 21.1 | 4.3 |
| FIa | 18 | 6.0 | 20.0 | 13.4 | 4.2 | 14 | 7.0 | 18.0 | 12.64 | 3.48 | 25 | 7.0 | 18.0 | 12.00 | 3.30 | 57 | 6.0 | 20.0 | 12.61 | 3.63 |
| FIp | 18 | 0.0 | 6.7 | 2.22 | 1.69 | 14 | 0.0 | 6.1 | 2.50 | 1.79 | 25 | 0.5 | 8.4 | 4.13 | 1.73 | 57 | 0.0 | 8.4 | 3.13 | 1.93 |
| FL | 18 | 75 | 128 | 104.3 | 11.5 | 14 | 95 | 121 | 105.7 | 6.8 | 25 | 100 | 136 | 112.3 | 8.8 | 57 | 75 | 136 | 108.2 | 9.9 |
| FuL | 17 | 44 | 60 | 51.1 | 4.6 | 15 | 39 | 53 | 47.8 | 5.0 | 25 | 30 | 59 | 46.5 | 7.8 | 57 | 30 | 60 | 48.2 | 6.5 |
| FW | 18 | 4.2 | 15.6 | 8.58 | 2.53 | 15 | 5.3 | 11.8 | 8.18 | 1.93 | 26 | 2.5 | 15.8 | 7.56 | 2.65 | 59 | 2.5 | 15.8 | 8.03 | 2.45 |
| HL | 18 | 56 | 78 | 67.2 | 6.0 | 15 | 55 | 78 | 63.9 | 7.2 | 26 | 55 | 71 | 62.2 | 4.2 | 59 | 55 | 78 | 64.2 | 6.0 |
| HW | 18 | 50 | 75 | 57.9 | 7.4 | 15 | 43 | 67 | 55.7 | 5.9 | 26 | 45 | 64 | 53.1 | 3.8 | 59 | 43 | 75 | 55.2 | 5.9 |
| PD-lt | 14 | 3.5 | 14.8 | 9.55 | 3.67 | 11 | 1.1 | 11.7 | 8.85 | 3.22 | 23 | 1.3 | 13.9 | 7.42 | 2.54 | 48 | 1.1 | 14.8 | 8.37 | 3.14 |
| PD-rt | 14 | 1.0 | 21.9 | 9.96 | 5.06 | 12 | 2.1 | 12.2 | 8.61 | 2.85 | 25 | 3.5 | 13.0 | 8.51 | 2.70 | 51 | 1.0 | 21.9 | 8.93 | 3.51 |
| PR-lt | 15 | 37 | 57 | 45.9 | 5.4 | 11 | 33 | 51 | 42.5 | 5.4 | 23 | 38 | 65 | 46.0 | 6.4 | 49 | 33 | 65 | 45.2 | 6.0 |
| PR-rt | 14 | 35 | 50 | 44.0 | 5.5 | 12 | 33 | 53 | 42.00 | 5.75 | 25 | 39 | 71 | 46.32 | 7.14 | 51 | 33 | 71 | 44.67 | 6.54 |
| Tcl | 16 | 13 | 25 | 17.3 | 3.1 | 12 | 11 | 22 | 17.1 | 2.7 | 21 | 13 | 22 | 18.0 | 2.5 | 49 | 11 | 25 | 17.5 | 2.7 |
| TL | 16 | 100 | 267 | 190.8 | 45.3 | 12 | 100 | 224 | 166.3 | 38.7 | 21 | 125 | 235 | 174.3 | 31.0 | 49 | 100 | 267 | 177.8 | 38.4 |

**Table A4.** Meristic counts for male *Sepia typica*. Sample size (*n*), minimum (Min), maximum (Max), arithmetic mean (Mean) and standard deviation of the mean (Std) are presented by region and for all regions combined for each character (See Table A1 for character abbreviations).

| Area | Northwest Area | | | | | Central Area | | | | | Eastern Area | | | | | Areas Combined | | | | |
|---|---|---|---|---|---|---|---|---|---|---|---|---|---|---|---|---|---|---|---|---|
| Char-acter | n | Min | Max | Mean | Std | n | Min | Max | Mean | Std | n | Min | Max | Mean | Std | n | Min | Max | Mean | Std |
| ASC1 | 20 | 29 | 44 | 36.2 | 3.9 | 22 | 28 | 42 | 35.2 | 3.4 | 22 | 30 | 41 | 35.6 | 3.2 | 64 | 28 | 44 | 35.6 | 3.5 |
| ASC1c | 21 | 5 | 7 | 6.1 | 0.4 | 22 | 5 | 8 | 6.3 | 0.7 | 22 | 4 | 8 | 6.4 | 1.1 | 65 | 4 | 8 | 6.3 | 0.8 |
| ASC1nt | 21 | 19 | 24 | 21.5 | 1.3 | 22 | 18 | 23 | 20.2 | 1.2 | 22 | 18 | 24 | 21 | 1.8 | 65 | 18 | 24 | 20.9 | 1.5 |
| ASC1p | 21 | 13 | 18 | 15.4 | 1.2 | 22 | 12 | 17 | 13.9 | 1.1 | 22 | 12 | 18 | 14.7 | 1.3 | 65 | 12 | 18 | 14.6 | 1.4 |
| ASC1t | 20 | 10 | 22 | 14.6 | 3.4 | 22 | 9 | 20 | 15.0 | 3.2 | 22 | 11 | 18 | 14.6 | 2.0 | 64 | 9 | 22 | 14.7 | 2.9 |
| ASC2 | 21 | 27 | 44 | 36.9 | 4.4 | 22 | 25 | 42 | 33.3 | 5.0 | 22 | 31 | 42 | 37.0 | 2.9 | 65 | 25 | 44 | 35.7 | 4.5 |
| ASC2c | 22 | 4 | 9 | 6.1 | 1.1 | 22 | 4 | 8 | 6.3 | 0.9 | 22 | 6 | 9 | 6.6 | 1.0 | 66 | 4 | 9 | 6.3 | 1.0 |
| ASC2nt | 22 | 20 | 26 | 22.7 | 1.8 | 22 | 19 | 24 | 21.1 | 1.5 | 22 | 20 | 26 | 22.1 | 1.6 | 66 | 19 | 26 | 22 | 1.7 |
| ASC2p | 22 | 14 | 19 | 16.5 | 1.3 | 22 | 13 | 17 | 14.8 | 1.1 | 22 | 13 | 18 | 15.5 | 1.2 | 66 | 13 | 19 | 15.6 | 1.4 |
| ASC2t | 21 | 5 | 21 | 14.1 | 3.9 | 22 | 4 | 19 | 12.2 | 4.5 | 22 | 10 | 20 | 14.9 | 2.5 | 65 | 4 | 21 | 13.7 | 3.8 |
| ASC3 | 22 | 29 | 49 | 42.2 | 4.3 | 22 | 32 | 48 | 40.6 | 4.6 | 22 | 37 | 48 | 41.7 | 2.9 | 66 | 29 | 49 | 41.5 | 4.0 |
| ASC3c | 21 | 6 | 8 | 6.3 | 0.6 | 22 | 4 | 8 | 6.3 | 0.9 | 20 | 4 | 8 | 6.4 | 0.9 | 63 | 4 | 8 | 6.3 | 0.8 |
| ASC3nt | 22 | 24 | 31 | 28.1 | 1.8 | 22 | 24 | 31 | 26.5 | 1.6 | 22 | 23 | 32 | 27.1 | 1.9 | 66 | 23 | 32 | 27.2 | 1.9 |
| ASC3p | 22 | 18 | 27 | 22.1 | 2.0 | 22 | 18 | 24 | 20.2 | 1.3 | 22 | 17 | 28 | 21.3 | 2.6 | 66 | 17 | 28 | 21.2 | 2.1 |

| Area | Northwest Area | | | | | Central Area | | | | | Eastern Area | | | | | Areas Combined | | | | |
|---|---|---|---|---|---|---|---|---|---|---|---|---|---|---|---|---|---|---|---|---|
| Character | n | Min | Max | Mean | Std | n | Min | Max | Mean | Std | n | Min | Max | Mean | Std | n | Min | Max | Mean | Std |
| ASC3t | 22 | 4 | 21 | 14.1 | 3.9 | 22 | 6 | 20 | 14.0 | 3.7 | 22 | 11 | 21 | 14.6 | 2.2 | 66 | 4 | 21 | 14.3 | 3.3 |
| ASC4 | 20 | 40 | 54 | 45.7 | 4.2 | 22 | 35 | 55 | 44.7 | 6.0 | 21 | 37 | 52 | 45.4 | 3.9 | 63 | 35 | 55 | 45.2 | 4.8 |
| ASC4c | 19 | 6 | 10 | 8.0 | 1.4 | 22 | 6 | 12 | 9.1 | 1.5 | 21 | 6 | 12 | 9.2 | 1.6 | 62 | 6 | 12 | 8.8 | 1.6 |
| ASC4m | 19 | 7 | 14 | 10.6 | 1.8 | 22 | 6 | 10 | 8.5 | 1.4 | 22 | 4 | 11 | 7.9 | 1.8 | 63 | 4 | 14 | 8.9 | 2.0 |
| ASC4nt | 19 | 27 | 34 | 30.8 | 1.9 | 22 | 25 | 34 | 30.1 | 2.6 | 21 | 26 | 36 | 30.4 | 2.5 | 62 | 25 | 36 | 30.4 | 2.4 |
| ASC4p | 19 | 10 | 14 | 12.2 | 1.3 | 22 | 10 | 15 | 12.5 | 1.5 | 22 | 11 | 16 | 13.1 | 1.3 | 63 | 10 | 16 | 12.6 | 1.4 |
| ASC4t | 19 | 10 | 22 | 15.1 | 3.7 | 22 | 3 | 22 | 14.6 | 4.9 | 22 | 11 | 22 | 15.1 | 2.7 | 63 | 3 | 22 | 14.9 | 3.8 |
| ClRC | 14 | 4 | 4 | 4.0 | 0.0 | 22 | 4 | 4 | 4.0 | 0.0 | 19 | 4 | 4 | 4.0 | 0.0 | 55 | 4 | 4 | 4.0 | 0.0 |
| CS# | 14 | 42 | 62 | 51.6 | 5.1 | 22 | 39 | 60 | 48.6 | 5.5 | 19 | 43 | 57 | 49.3 | 3.6 | 55 | 39 | 62 | 49.6 | 4.9 |
| Hect | 21 | 40 | 54 | 46.1 | 3.6 | 22 | 30 | 53 | 43.1 | 5.8 | 22 | 40 | 53 | 47.4 | 3.5 | 65 | 30 | 54 | 45.6 | 4.7 |
| HSCm | 21 | 18 | 23 | 21.2 | 1.6 | 22 | 17 | 23 | 20.5 | 1.5 | 22 | 18 | 26 | 21.4 | 1.9 | 65 | 17 | 26 | 21.0 | 1.7 |
| HSCt | 21 | 10 | 22 | 15.2 | 3.1 | 22 | 2 | 21 | 13.7 | 5.5 | 22 | 13 | 21 | 16.6 | 2.2 | 65 | 2 | 22 | 15.2 | 4.0 |
| HSCu | 21 | 8 | 13 | 9.8 | 1.6 | 22 | 6 | 11 | 8.9 | 1.2 | 22 | 7 | 13 | 9.4 | 1.2 | 65 | 6 | 13 | 9.4 | 1.4 |
| P#-lt | 21 | 10 | 13 | 11.6 | 1.2 | 22 | 9 | 13 | 10.7 | 1.1 | 20 | 8 | 13 | 11.1 | 1.5 | 63 | 8 | 13 | 11.1 | 1.3 |
| P#-rt | 22 | 9 | 15 | 11.5 | 1.7 | 23 | 9 | 13 | 10.6 | 1.1 | 21 | 8 | 13 | 10.6 | 1.2 | 66 | 8 | 15 | 10.9 | 1.4 |
| TrRC | 13 | 12 | 15 | 14.1 | 1.0 | 22 | 11 | 15 | 13.5 | 1.2 | 19 | 12 | 15 | 13.7 | 1.0 | 55 | 11 | 15 | 13.7 | 1.1 |

**Table A5.** Meristic counts for female *Sepia typica*. Sample size (*n*), minimum (Min), maximum (Max), arithmetic mean (Mean) and standard deviation of the mean (Std) are presented by region and for all regions combined for each character (See Table A1 for character abbreviations).

| Area | Northwest Area | | | | | Central Area | | | | | Eastern Area | | | | | Areas Combined | | | | |
|---|---|---|---|---|---|---|---|---|---|---|---|---|---|---|---|---|---|---|---|---|
| Character | n | Min | Max | Mean | Std | n | Min | Max | Mean | Std | n | Min | Max | Mean | Std | n | Min | Max | Mean | Std |
| ASC1 | 18 | 32 | 52 | 43.2 | 4.9 | 15 | 34 | 49 | 41.4 | 4.1 | 26 | 35 | 52 | 41.6 | 4.4 | 59 | 32 | 52 | 42.0 | 4.5 |
| ASC1nt | 18 | 21 | 34 | 29.4 | 3.4 | 15 | 24 | 36 | 28.9 | 3.5 | 26 | 21 | 37 | 29.2 | 3.8 | 59 | 21 | 37 | 29.2 | 3.5 |
| ASC1t | 18 | 10 | 20 | 13.8 | 2.4 | 15 | 10 | 16 | 12.5 | 1.8 | 26 | 6 | 20 | 12.5 | 2.8 | 59 | 6 | 20 | 12.9 | 2.5 |
| ASC2 | 18 | 33 | 56 | 44.8 | 5.0 | 15 | 33 | 47 | 42.3 | 3.8 | 26 | 35 | 54 | 44.2 | 4.3 | 59 | 33 | 56 | 43.9 | 4.4 |
| ASC2nt | 18 | 23 | 37 | 30.4 | 3.6 | 15 | 24 | 34 | 28.9 | 2.9 | 26 | 24 | 38 | 31.0 | 3.2 | 59 | 23 | 38 | 30.3 | 3.3 |
| ASC2t | 18 | 10 | 19 | 14.4 | 2.8 | 15 | 9 | 18 | 13.5 | 2.6 | 26 | 9 | 17 | 13.2 | 2.2 | 59 | 9 | 19 | 13.7 | 2.5 |
| ASC3 | 18 | 34 | 60 | 50.1 | 6.1 | 15 | 41 | 53 | 47.2 | 3.9 | 26 | 45 | 59 | 50.5 | 4.1 | 59 | 34 | 60 | 49.5 | 4.9 |
| ASC3nt | 18 | 27 | 44 | 34.4 | 4.4 | 15 | 24 | 41 | 32.1 | 4.1 | 26 | 30 | 42 | 35.9 | 3.0 | 59 | 24 | 44 | 34.5 | 4.0 |
| ASC3t | 18 | 7 | 21 | 15.6 | 4.0 | 15 | 12 | 19 | 15.1 | 2.1 | 26 | 11 | 21 | 14.6 | 2.4 | 59 | 7 | 21 | 15.0 | 2.9 |
| ASC4 | 18 | 42 | 59 | 50.1 | 4.6 | 15 | 44 | 56 | 49.3 | 3.7 | 26 | 42 | 59 | 50.9 | 5.1 | 59 | 42 | 59 | 50.3 | 4.6 |
| ASC4nt | 18 | 30 | 39 | 34.4 | 2.6 | 15 | 29 | 41 | 32.7 | 2.9 | 26 | 28 | 43 | 34.7 | 3.2 | 59 | 28 | 43 | 34.1 | 3.0 |
| ASC4t | 18 | 7 | 21 | 15.7 | 3.8 | 15 | 13 | 22 | 16.7 | 2.6 | 26 | 9 | 23 | 16.2 | 3.6 | 59 | 7 | 23 | 16.2 | 3.4 |
| ClRC | 16 | 4 | 4 | 4.0 | 0.0 | 12 | 4 | 4 | 4.0 | 0.0 | 21 | 4 | 4 | 4.0 | 0.0 | 49 | 4 | 4 | 4.0 | 0.0 |
| CS# | 16 | 43 | 58 | 52.4 | 4.4 | 12 | 42 | 54 | 48.0 | 3.3 | 21 | 47 | 63 | 52.3 | 3.8 | 49 | 42 | 63 | 51.3 | 4.3 |
| P#-lt | 16 | 9 | 14 | 11.8 | 1.4 | 11 | 8 | 13 | 11.4 | 1.5 | 23 | 8 | 13 | 11.0 | 1.5 | 50 | 8 | 14 | 11.3 | 1.5 |
| P#-rt | 15 | 10 | 15 | 11.9 | 1.2 | 13 | 9 | 15 | 11.1 | 1.5 | 25 | 9 | 14 | 11.6 | 1.3 | 53 | 9 | 15 | 11.5 | 1.3 |
| TrRC | 16 | 12 | 16 | 13.8 | 1.1 | 12 | 12 | 14 | 13.2 | 0.6 | 21 | 13 | 15 | 13.9 | 0.7 | 49 | 12 | 16 | 14.0 | 0.9 |

**Table A6.** Univariate regression equations (Y = a + bML) used to calculate residuals, where *a* = intercept, *b* = slope, ML = mantle length, $r^2$ = correlation coefficient of the regression, and *Y* = the predicted value of that dependent variable. Sig. is the significance of the regression model (***, $p < 0.001$; **, $p < 0.01$; *, $p < 0.5$; NS, not significant).

| Y | a | b | $r^2$ | Sig. | Y | a | b | r | Sig. | Y | a | b | r | Sig. |
|---|---|---|---|---|---|---|---|---|---|---|---|---|---|---|
| **Females** | | | | | **Males** | | | | | **Males (continued)** | | | | |
| AL 1 | 0.99 | 0.41 | 0.51 | *** | AL 1 | 2.75 | 0.34 | 0.39 | *** | AS4ps | 0.05 | 0.02 | 0.26 | *** |
| AL 2 | 0.04 | 0.45 | 0.67 | *** | AL 2 | 1.65 | 0.43 | 0.56 | *** | ASl4 | 0.05 | 0.03 | 0.42 | *** |
| AL 3 | 1.52 | 0.38 | 0.59 | *** | AL 3 | 0.84 | 0.52 | 0.68 | *** | ASl4m | 0.17 | 0.00 | 0.02 | NS |
| AL 4 | 2.9 | 0.38 | 0.40 | *** | AL 4 | 2.59 | 0.49 | 0.32 | *** | FFu | 2.02 | 0.09 | 0.06 | * |
| AS1pl | 0.33 | 0.01 | 0.22 | *** | AMH | −0.57 | 0.07 | 0.16 | ** | FIa | 0.71 | 0.07 | 0.08 | * |
| AS2pl | 0.25 | 0.02 | 0.39 | *** | AS1c | 0.29 | 0.02 | 0.35 | *** | FIp | 0 | 0.03 | 0.05 | NS |
| AS3pl | 0.27 | 0.02 | 0.30 | *** | AS1pl | 0.19 | 0.02 | 0.47 | *** | FL | 1.63 | 0.97 | 0.62 | *** |
| AS4pl | 0.19 | 0.02 | 0.52 | *** | AS1ps | 0.19 | 0.02 | 0.21 | *** | FuL | 1.89 | 0.38 | 0.38 | *** |
| FFu | 0.23 | 0.20 | 0.31 | *** | AS2c | 0.17 | 0.04 | 0.47 | *** | FW | −1.15 | 0.14 | 0.29 | *** |
| FIa | 0.83 | 0.08 | 0.10 | * | AS2pl | 0.23 | 0.03 | 0.47 | *** | HcL | 1.21 | 0.52 | 0.53 | *** |
| FIp | 0.23 | 0.03 | 0.01 | NS | AS2ps | 0.17 | 0.02 | 0.35 | *** | HL | 3.73 | 0.44 | 0.58 | *** |
| FL | 1.06 | 1.03 | 0.74 | *** | AS3c | 0.08 | 0.04 | 0.59 | *** | HW | 3.14 | 0.38 | 0.60 | *** |
| FuL | 3.75 | 0.28 | 0.33 | *** | AS3pl | 0.22 | 0.03 | 0.42 | *** | MHL | 1.6 | 0.26 | 0.34 | *** |
| FW | −0.06 | 0.08 | 0.20 | *** | AS3ps | 0.09 | 0.03 | 0.38 | *** | MLv | 2.42 | 0.77 | 0.75 | *** |
| HL | 2.31 | 0.51 | 0.60 | *** | AS4c | 0.04 | 0.03 | 0.42 | *** | PD-lt | 0.69 | 0.04 | 0.03 | NS |
| HW | 3.58 | 0.36 | 0.57 | *** | AS4pl | 0.08 | 0.04 | 0.45 | *** | PR-lt | 1.66 | 0.34 | 0.37 | *** |
| MLv | −0.45 | 0.93 | 0.82 | *** | | | | | | | | | | |

## Appendix B

**Material Examined**: 141 specimens (75 Male and 66 Female):

Two specimens (2 Female: SAIAB 211521—ML 19 mm TW 2.8 g., SAIAB 211522—ML 18 mm TW 2.6 g.) FRS Africana voyage Afr−050, 14 January 1987, Station A05254, 30°50′00.0″ S, 16°18′00.0″ E, bottom trawl 268 m; SAIAB 211523—Male: ML 18 mm TW 2.0 g. FRS Africana voyage Afr−095, 14 September 1991, Station A12002, 35°04′00.0″ S, 20°21′00.0″ E, bottom trawl 105 m; SAIAB 211524—Male: ML 17 mm TW 0.9 g. FRS Africana voyage Afr-095, 29 September 1991, Station A12097, 34°04′00.0″ S, 26°33′00.0″ E, bottom trawl 112 m; SAIAB 211525—Female: ML 29 mm TW 6.6 g. FRS Africana voyage Afr-102, 13 April 1992, Station A13420, 34°20′00.0″ S, 22°08′00.0″ E, bottom trawl 68 m; 2 Specimens (2 Male: SAIAB 211526—ML 20 mm TW 2.9 g., SAIAB 211527—ML 18 mm TW 1.7 g.) FRS Africana voyage Afr-106, 15 September 1992, Station A13946, 34°07′00.0″ S, 23°37′00.0″ E, bottom trawl 87 m; SAIAB 211528—Male: ML 21 mm TW 2.3 g. FRS Africana voyage Afr-182, 13 September 2003, Station A21995, 33°49′00.0″ S, 26°33′00.0″ E, bottom trawl 50 m; 5 Specimens (3 Male: SAIAB 211529—ML 23 mm TW 3.1 g., SAIAB 211530—ML 19 mm TW 2.0 g., SAIAB 211531—ML 20 mm TW 2.6 g.; 2 Female: SAIAB 211532—ML 19 mm TW 2.6 g., SAIAB 211533—ML 22 mm TW 3.2 g.) FRS Africana voyage Afr-224, 16 September 2006, Station A26487, 34°39′00.0″ S, 20°34′00.0″ E, bottom trawl 69 m; SAIAB 211534—Male: ML 23 mm TW 4.5 g. FRS Africana voyage Afr-249, 06 February 2009, Station A29495, 30°32′00.0″ S, 16°03′00.0″ E, bottom trawl 255 m; SAIAB 211535—Male: ML 22 mm TW 3.0 g. FRS Africana voyage Afr-273, 06 May 2011, Station A31668, 35°25′49.1″ S, 21°55′00.0″ E, bottom trawl 140 m; SAIAB 211536—Male: ML 21 mm TW 2.7 g. R/V Dr Fridjof Nansen voyage 2000401, 04 February 2000, Station AN0101, 31°01′00.0″ S, 16°43′00.0″ E, bottom trawl 248 m; SAIAB 211537—Female: ML 20 mm TW 3.7 g. R/V Dr Fridjof Nansen voyage 2000401, 07 February 2000, Station AN0112, 30°58′00.0″ S, 16°14′00.0″ E, bottom trawl 281 m; SAIAB 211538—Male: ML 16 mm TW 1.8 g. R/V Dr Fridjof Nansen voyage 2000401, 09 February 2000, Station AN0127, 30°12′00.0″ S, 16°02′00.0″ E, bottom trawl 206 m; 2 Specimens (2 Male: SAIAB 211539—ML 19 mm TW 2.9 g., SAIAB 211540—ML 22 mm TW 3.1 g.) R/V Dr Fridjof Nansen voyage 2000401, 9 February 2000, Station AN0128, 30°08′00.0″ S, 16°00′00.0″ E, bottom trawl 200 m; 3 Specimens (3 Male: SAIAB 211541—ML

22 mm TW 3.5 g., SAIAB 211542—ML 17 mm TW 1.4 g., SAIAB 211543—ML 18 mm TW 2.0 g.) R/V Dr Fridjof Nansen voyage 2000404, 20 May 2000, Station AN0149, 35°39′00.0″ S, 20°32′00.0″ E, bottom trawl 137 m; SAIAB 211544—Female: ML 18 mm TW 2.3 g. R/V Dr Fridjof Nansen voyage 2000404, 20 May 2000, Station AN0151, 35°47′00.0″ S, 20°57′00.0″ E, bottom trawl 114 m; 5 Specimens (2 Male: SAIAB 211545—ML 18 mm TW 2.1 g., SAIAB 211546—ML 17 mm TW 1.8 g.; 3 Female: SAIAB 211547—ML 12 mm TW 0.9 g., SAIAB 211548—ML 19 mm TW 2.6 g., SAIAB 211549—ML 20 mm TW 2.7 g.) R/V Dr Fridjof Nansen voyage 2000404, 21 May 2000, Station AN0153, 36°33′00.0″ S, 20°31′00.0″ E, bottom trawl 182 m; SAIAB 211550—Female: ML 17 mm TW 2.0 g. R/V Dr Fridjof Nansen voyage 2000404, 21 May 2000, Station AN0154, 36°36′00.0″ S, 20°36′00.0″ E, bottom trawl 183 m; SAIAB 211551—Male: ML 18 mm TW 1.6 g. R/V Dr Fridjof Nansen voyage 2000404, 26 May 2000, Station AN0171, 35°00′00.0″ S, 22°59′00.0″ E, bottom trawl 202 m; 2 Specimens (1 Male: SAIAB 211552—ML 20 mm TW 2.4 g.; 1 Female: SAIAB 211553—ML 23 mm TW 4.5 g.) R/V Dr Fridjof Nansen voyage 2000404, 28 May 2000, Station AN0185, 34°02′00.0″ S, 25°09′00.0″ E, bottom trawl 49 m; 14 Specimens (7 Male: SAIAB 211554—ML 21 mm TW 3.1 g., SAIAB 211555—ML 18 mm TW 2.2 g., SAIAB 211556—ML 15 mm TW 1.5 g., SAIAB 211557—ML 20 mm TW 3.1 g., SAIAB 211558—ML 19 mm TW 2.7 g., SAIAB 211559—ML 15 mm TW 1.5 g., SAIAB 211560—ML 17 mm TW 2.6 g.; 7 Female: SAIAB 211561—ML 24 mm TW 4.4 g., SAIAB 211562—ML 16 mm TW 1.6 g., SAIAB 211563—ML 17 mm TW 2.2 g., SAIAB 211564—ML 19 mm TW 2.5 g., SAIAB 211565—ML 18 mm TW 2.4 g., SAIAB 211566—ML 17 mm TW 1.9 g., SAIAB 211567—ML 20 mm TW 3.8 g.) R/V Dr Fridjof Nansen voyage 2000404, 30 May 2000, Station AN0193, 33°53′00.0″ S, 26°16′00.0″ E, bottom trawl 50 m; 3 Specimens (1 Male: SAIAB 211568—ML 16 mm TW 1.9 g.; 2 Female: SAIAB 211569—ML 19 mm TW 2.7 g., SAIAB 211570—ML 20 mm TW 3.5 g.) R/V Dr Fridjof Nansen voyage 2000404, 30 May 2000, Station AN0194, 33°50′00.0″ S, 26°33′00.0″ E, bottom trawl 50 m; 8 Specimens (2 Male: SAIAB 211571—ML 18 mm TW 2.9 g., SAIAB 211572—ML 19 mm TW 3.0 g.; 6 Female: SAIAB 211573—ML 22 mm TW 3.9 g., SAIAB 211574—ML 20 mm TW 3.5 g., SAIAB 211575—ML 20 mm TW 3.2 g., SAIAB 211577—ML 16 mm TW 1.7 g., SAIAB 211578—ML 20 mm TW 3.8 g., SAIAB 211579—ML 20 mm TW 2.9 g.) R/V Dr Fridjof Nansen voyage 2000404, 30 May 2000, Station AN0195, 33°44′00.0″ S, 26°48′00.0″ E, bottom trawl 68 m; 20 Specimens (10 Male: SAIAB 211580—ML 21 mm TW 3.7 g., SAIAB 211581—ML 19 mm TW 3.1 g., SAIAB 211582—ML 18 mm TW 2.2 g., SAIAB 211583—ML 17 mm TW 1.9 g., SAIAB 211584—ML 12 mm TW 0.6 g., SAIAB 211585—ML 19 mm TW 2.5 g., SAIAB 211586—ML 22 mm TW 3.6 g., SAIAB 211587—ML 19 mm TW 2.4 g., SAIAB 211588—ML 20 mm TW 2.9 g., SAIAB 211589—ML 13 mm TW 1.0 g.; 10 Female: SAIAB 211590—ML 20 mm TW 3.3 g., SAIAB 211591—ML 20 mm TW 3.4 g., SAIAB 211592—ML 19 mm TW 2.1 g., SAIAB 211593—ML 18 mm TW 2.5 g., SAIAB 211594—ML 19 mm TW 3.5 g., SAIAB 211595—ML 16 mm TW 1.6 g., SAIAB 211596—ML 20 mm TW 2.9 g., SAIAB 211597—ML 14 mm TW 1.3 g., SAIAB 211598—ML 20 mm TW 3.2 g., SAIAB 211599—ML 22 mm TW 3.4 g.) R/V Dr Fridjof Nansen voyage 2000404, 30 May 2000, Station AN0196, 33°54′00.0″ S, 26°36′00.0″ E, bottom trawl 105 m; SAIAB 211600—Female: ML 19 mm TW 3.0 g. R/V Dr Fridjof Nansen voyage 2000404, 05 June 2000, Station AN0213, 34°27′00.0″ S, 22°18′00.0″ E, bottom trawl 88 m; 2 Specimens (2 Male: SAIAB 211601—ML 19 mm, SAIAB 211602—ML 19 mm TW 1.9 g.) R/V Dr Fridjof Nansen voyage 2000404, 8 June 2000, Station AN0228, 34°34′00.0″ S, 21°33′00.0″ E, bottom trawl 71 m; 4 Specimens (2 Male: SAIAB 211603—ML 21 mm TW 2.4 g., SAIAB 211604—ML 18 mm TW 2.0 g.; 2 Female: SAIAB 211605—ML 23 mm TW 2.7 g., SAIAB 211606—ML 19 mm TW 1.9 g.) R/V Dr Fridjof Nansen voyage 2000404, 9 June 2000, Station AN0233, 34°56′00.0″ S, 20°56′00.0″ E, bottom trawl 90 m; 6 Specimens (5 Male: SAIAB 211607—ML 22 mm TW 3.0 g., SAIAB 211608—ML 17 mm TW 1.8 g., SAIAB 211609—ML 17 mm TW 2.2 g., SAIAB 211610—ML 20 mm TW 2.1 g., SAIAB 211611—ML 17 mm TW 1.6 g.; 1 Female: SAIAB 211612—ML 17 mm TW 2.2 g.) R/V Dr Fridjof Nansen voyage 2000404, 10 June 2000, Station AN0234, 34°55′00.0″ S, 20°28′00.0″ E, bottom trawl 106 m; 12 Specimens (7 Male: SAIAB 211613—ML 21 mm TW 2.6 g., SAIAB 211614—ML 17 mm TW 1.8 g., SAIAB 211615—ML 17 mm TW 1.8 g., SAIAB

211616—ML 20 mm TW 2.5 g., SAIAB 211617—ML 23 mm TW 3.3 g., SAIAB 211618—ML 18 mm TW 2.0 g., SAIAB 211619—ML 20 mm TW 2.7 g.; 5 Female: SAIAB 211620—ML 19 mm TW 2.4 g., SAIAB 211621—ML 17 mm TW 2.3 g., SAIAB 211622—ML 15 mm TW 1.7 g., SAIAB 211623—ML 22 mm TW 4.1 g., SAIAB 211624—ML 15 mm TW 1.5 g.) R/V Dr Fridjof Nansen voyage 2000404, 10 June 2000, Station AN0236, 35°01′00.0″ S, 20°10′00.0″ E, bottom trawl 104 m; SAIAB 211625—Male: ML 18 mm TW 2.8 g. R/V Dr Fridjof Nansen voyage 2003401, 20 January 2003, Station AN0560, 31°12′00.0″ S, 16°27′00.0″ E, bottom trawl 313 m; SAIAB 211626—Female: ML 25 mm TW 4.7 g. R/V Dr Fridjof Nansen voyage 2003401, 23 January 2003, Station AN0576, 30°24′00.0″ S, 16°33′00.0″ E, bottom trawl 214 m; 3 Specimens (1 Male: SAIAB 211627—ML 18 mm TW 1.8 g.; 2 Female: SAIAB 211628—ML 23 mm TW 3.8 g., SAIAB 211629—ML 18 mm TW 3.1 g.) R/V Dr Fridjof Nansen voyage 2003401, 23 January 2003, Station AN0577, 30°33′00.0″ S, 16°18′00.0″ E, bottom trawl 245 m; 3 Specimens (1 Male: SAIAB 211630—ML 22 mm TW 3.4 g.; 2 Female: SAIAB 211631—ML 22 mm TW 3.3 g., SAIAB 211632—ML 23 mm TW 4.0 g.) R/V Dr Fridjof Nansen voyage 2003401, 23 January 2003, Station AN0578, 30°41′00.0″ S, 16°03′00.0″ E, bottom trawl 208 m; SAIAB 211633—Female: ML 24 mm TW 4.7 g. R/V Dr Fridjof Nansen voyage 2003401, 24 January 2003, Station AN0582, 30°07′00.0″ S, 16°20′00.0″ E, bottom trawl 187 m; 3 Specimens (1 Male: SAIAB 211634—ML 13 mm TW 0.9 g.; 2 Female: SAIAB 211635—ML 12 mm TW 0.8 g., SAIAB 211636—ML 21 mm TW 2.6 g.) R/V Dr Fridjof Nansen voyage 2003401, 25 January 2003, Station AN0587, 29°57′00.0″ S, 15°54′00.0″ E, bottom trawl 200 m; SAIAB 211637—Male: ML 17 mm TW 2.2 g. R/V Dr Fridjof Nansen voyage 2006402, 22 February 2006, Station AN1271, 30°55′58.8″ S, 16°25′58.8″ E, bottom trawl 276 m; SAIAB 211638—Female: ML 16 mm TW 1.5 g. R/V Dr Fridjof Nansen voyage 2007401, 30 January 2007, Station Aborted trawl, 30°59′36.0″ S, 16°15′12.0″ E, bottom trawl 282 m; 4 Specimens (3 Male: SAIAB 211639—ML 22 mm TW 3.1 g., SAIAB 211640—ML 19 mm TW 2.3 g., SAIAB 211641—ML 22 mm TW 2.7 g.; 1 Female: SAIAB 211642—ML 16 mm TW 1.8 g.) R/V Dr Fridjof Nansen voyage 2007401, 29 January 2007, Station AN1361, 31°12′16.8″ S, 16°27′47.4″ E, bottom trawl 319 m; 8 Specimens (5 Male: SAIAB 211643—ML 18 mm TW 2.3 g., SAIAB 211644—ML 17 mm TW 2.0 g., SAIAB 211645—ML 23 mm TW 3.2 g., SAIAB 211646—ML 18 mm TW 2.0 g., SAIAB 211647—ML 21 mm TW 3.2 g.; 3 Female: SAIAB 211648—ML 19 mm TW 3.5 g., SAIAB 211649—ML 20 mm TW 3.5 g., SAIAB 211650—ML 20 mm TW 3.1 g.) R/V Dr Fridjof Nansen voyage 2007401, 30 January 2007, Station AN1364, 31°00′07.2″ S, 16°15′22.8″ E, bottom trawl 286 m; 3 Specimens (2 Male: SAIAB 211651—ML 18 mm TW 2.6 g., SAIAB 211652—ML 20 mm TW 2.6 g.; 1 Female: SAIAB 211653—ML 17 mm TW 2.2 g.) R/V Dr Fridjof Nansen voyage 2009401, 7 February 2009, Station AN1705, 29°48′57.6″ S, 15°13′30.0″ E, bottom trawl 231 m; SAIAB 211654—Female: ML 21 mm R/V Dr Fridjof Nansen voyage 2010401, 11 January 2010, Station AN1770, 35°57′27.6″ S, 20°43′50.4″ E, bottom trawl 139 m; 2 Specimens (1 Male: SAIAB 211655—ML 21 mm; 1 Female: SAIAB 211656—ML 19 mm TW 2.1 g.) R/V Dr Fridjof Nansen voyage 2010401, 12 January 2010, Station AN1775, 34°39′43.8″ S, 20°31′23.4″ E, bottom trawl 71 m; 2 Specimens (2 Female: SAIAB 211657—ML 18 mm TW 2.4 g., SAIAB 211658—ML 21 mm TW 3.8 g.) R/V Dr Fridjof Nansen voyage 2012401, 15 February 2012, Station AN2240, 30°36′20.4″ S, 16°52′04.2″ E, bottom trawl 193 m; 2 Specimens (1 Male: SAIAB 211659—ML 25 mm; 1 Female: SAIAB 211660—ML 23 mm TW 5.1 g.) R/V Dr Fridjof Nansen voyage 2012401, 18 February 2012, Station AN2256, 30°08′59.4″ S, 16°20′53.4″ E, bottom trawl 190 m; SAIAB 211661—Male: ML 22 mm TW 3.9 g. R/V Dr Fridjof Nansen voyage 2012401, 18 February 2012, Station AN2257, 30°00′55.2″ S, 16°30′33.0″ E, bottom trawl 179 m; SAIAB 211684—Female: ML 23 mm TW 4.4 g. R/V Dr Fridjof Nansen voyage 2004405, 24 April 2004, Station, 29°22′01.2″ S, 14°37′58.8″ E, bottom trawl 326 m.

**Appendix C**

Images of the holotype of *Sepia typica*, NHMD-77360.

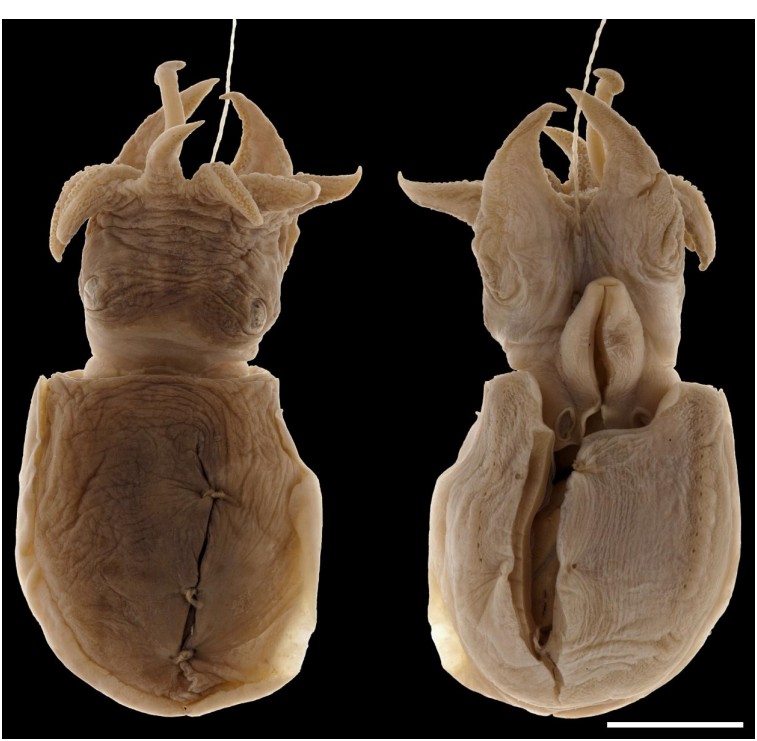

**Figure A1.** *Sepia typica* (Steenstrup, 1875) holotype (NHMD-77360, female ML 26, 6gm, Zoological Museum, Copenhagen), dorsal (**left**) and ventral (**right**) aspects. Scale bar 10 mm.

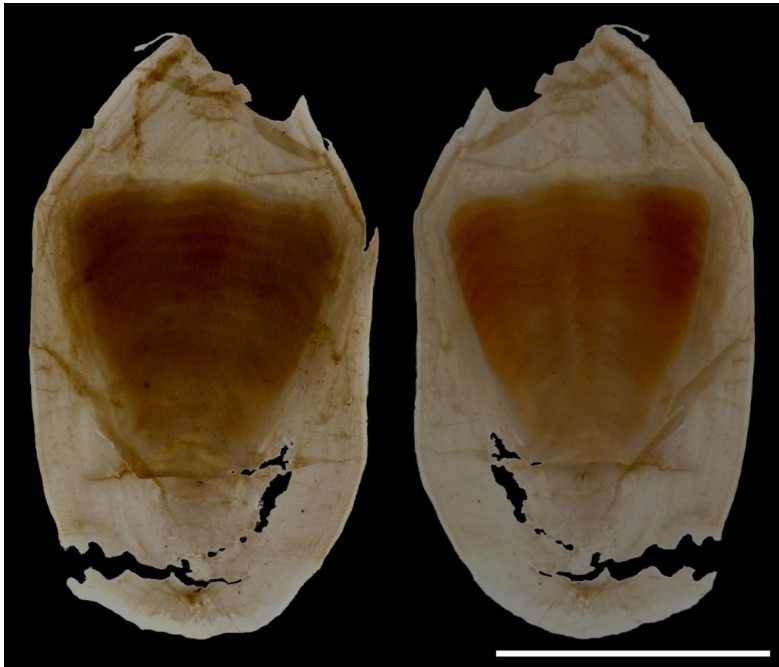

**Figure A2.** Dorsal (**left**) and ventral (**right**) aspects of the cuttlebone of the holotype of *Sepia typica* (Steenstrup, 1875) (NHMD-77360, female ML 26, 6gm, Zoological Museum, Copenhagen). Scale bar 10 mm.

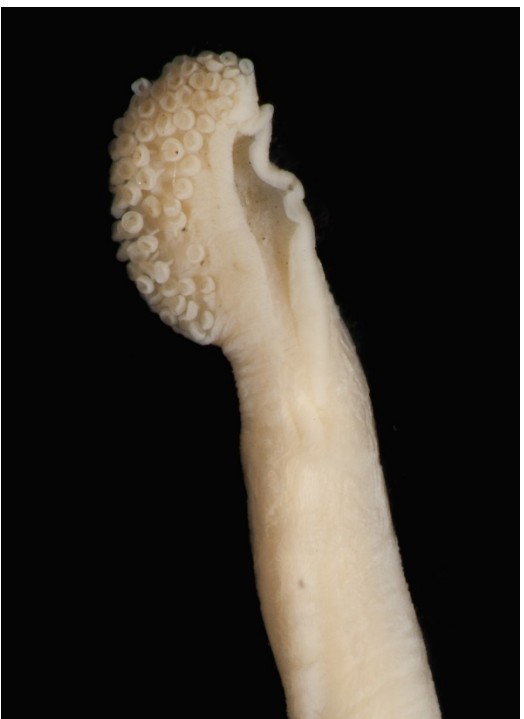

**Figure A3.** Tentacular club of the holotype of *Sepia typica* (Steenstrup, 1875) (NHMD-77360, female ML 26, 6gm, Zoological Museum, Copenhagen).

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
