# Peer review of "Detailed Description and Morphological Assessment of Sepia typica (Steenstrup, 1875) (Cephalopoda: Sepiidae)"

_diversity, doi:10.3390/d14121073_

Round 1

Reviewer 1 Report

This is a nice study, once again underlining the importance of ‘good old’ taxonomy. I would only suggest a couple of minor changes and corrections.

Lines 64-65: please provide it all as supplementary information for this paper. It wouldn’t take much time to prepare, but would significantly benefit our research community. Also, it would be in line with currently applied FAIR data approach.

Figure 14 and Figure 19: please provide the photographs, or better drawings. This drawing quality drags the paper down, sorry for saying that. I might have attached papers with an example drawings or photos of the suckers, radulae and spermatophores, but I’m sure you have a lot of these papers. Particular comments are: Fig. 14. Suckers, it’s better to set them in equal size, and give each a scale, as now it’s impossible to see the teeth on the smaller sucker; Fig. 19a. Spermatophore, doesn’t give the information on proportions of the cement body and ejaculatory apparatus, which are highly variable in Sepia; Fig. 19b. Radula, REM photo would be much more informative, if that’s possible to provide.

Reviewer 2 Report

Reviewer comments. Redescription and Systematic Assessment of Sepia typica (Steenstrup, 1875) (Cephalopoda: Sepiidae)

This is a detailed, well written and well-illustrated morphological account of a relatively poorly known cephalopod. It will serve as a very valuable baseline for future research, in particular further testing of the ‘single-species’ hypothesis using molecular tools when fresh specimens become available from across the considerable range of the taxon. It would be interesting to examine its population genetics to perhaps underpin the significant morphological variation that has been reported.

I have some slight concerns pertaining to the statistical analysis and suggest reanalysis/reexamination of the morphometric data using different and additional methodologies to perhaps obtain further insights into the variability (or lack thereof) that has been noted; at the very least to determine whether this gives the same result.

To this end I recommend testing the methodology outlined in a paper by Winterbottom et al. (1984) Geographic variation in Congrogadus subducens (Teleostei, Perciformes, Congrogadidae). Can. J. Zool. 62: 1605.

For any statistical analysis of morphometric specimen data, the effect of size variation between specimens needs to be minimized to enable comparison to be made based on shape. This will address the problem of allometry. Therefore, a procedure that removes or reduces size influences is needed when using morphometric characters to discriminate among animals that vary in size. I am not certain that this factor has been properly addressed in the present analyses. A method for dealing with this is outlined in Winterbottom et al. (1984) and was used by Reid (1991: 753) in a study of geographic variation in Australian Rossiinae. (Reid A (1991) Taxonomic review of the Australian Rossiinae (Cephalopoda: Sepiolidae), with a description of a new species, Neorossia leptodons, and a redescription of N. caroli (Joubin, 1902). Bull. Mar. Sci. 49(3): 748-831.)

This method involves using residual variables as a method of size removal. These values can then be interpreted as size and allometry-free shape estimators.

In addition to the broader analyses, using this method may lend clarity to the data presented in 3.3 (page 6 of the manuscript) that examines geographic variation in male arm morphology. Using ratios to compare the diameters of the smallest proximal sucker to the largest cluster sucker as a proportion of mantle length gives a size-free (but not an allometry free) shape estimator. However, there are several statistical and conceptual problems in using ratios for statistical comparisons among organisms. Primarily these are that the ratio can behave spuriously depending on the correlation of the numerator and the denominator, and the ratio is not necessarily independent of the denominator.

It would also be informative to apply latitude and longitudinal regression analyses to meristic variables and morphometric PCA scores and residuals to examine geographic variation in this taxon as outlined in Reid (1991: 756). Distinct populations detected by these techniques could reflect ecophenotypic variation, genotypic variation or both. If variation reflects genetic differences and therefore population groups, particular characters may react in a different way to an environmental gradient.

Further analyses of the morphometric and meristic data is recommended to obtain additional insights into the variability that has been observed in this species and whether this might be explained by clinal variation.

Detailed comments for consideration by the authors follow. Numbers refer to text line numbers.

2-3: Title. The title is somewhat misleading. Redescription of a taxon is normally based on types and additional material. However, while figures of the holotype are provided in A1-A3, the actual type specimen was not examined. Borrowing types is not always possible. If attempts were made, this should be mentioned, or a clear reason given regarding why the type specimen was not borrowed and examined. In addition, it would be useful to indicate the type locality (albeit rather general ‘off Table Bay’) in Figure 1. As no detailed collection locality was provided in the original description it would be helpful to indicate the location of Table Bay so that it is clear to readers that the specimens examined in detail in this study clearly encompass the original type locality.

‘Systematic Assessment’ is also not strictly correct in that systematics refers to the evolutionary relationships among taxa. However, this paper refers to the single taxon Sepia typica and does not explore its relationship to other taxa. A more realistic title could be ‘Complete Description and Morphological Assessment of Sepia typica from South Africa’.

Note in the title line 2 ‘sepia’ should be capitalised.

Line 3: steenstrup, cephalopoda and sepiidae should be capitalised.

Why is the genus name Hemisepius not used as it has been used in previous published studies by the authors? This requires explanation in the Introduction.

12: redescribed (not re-described)

12-13: Morphometric and meristic characters are morphological characters. Suggest reword to ‘Analyses of qualitative and quantitative morphological characters strongly suggest that . . . ‘

(In addition, it is grammatically incorrect to state, ‘Its morphological, morphometric and meristic characters strongly suggest’. . .. These characters are not capable of suggesting anything; it is the examination/analysis of these characters that leads to the suggestion that S. typica is a single species.)

16: delete the word ‘either’

27: remove semi-colon

31: It is not clear what ‘More specifically’ refers to. This requires explanation.

33: 50 needs a closing bracket

40: ‘she considered’ – do you mean ‘she examined’?

51: taxonomic status not ‘systematic status’ – the latter referring to phylogenetic relationships among taxa

54: were, rather than ‘have been’

56: Was the formalin buffered? If not, this might explain (in part) decalcification of cuttlebones.

66: Indicate Table Bay on both maps and in caption add ‘Table Bay is the type locality of S. typica’.

Indicate on horizontal and vertical axes °E and °S respectively.

a)       S. typica males (plural) and b) females (plural)

69–70: edit to ‘Orange line = coastline, black line = 500 m isobath.’

74: add comma after However

81: replace ‘which’ with ‘that’ (here and elsewhere if ‘that’ can be used)

81-83: a better definition for ClCR is ‘number of suckers in oblique transverse rows’, and TrRC ‘number of oblique transverse rows of suckers’.

88: correct spelling from mantel to ‘mantle’

95: All other measurements ‘refer’ to the nearest mm. (Otherwise sentence is incomplete.)

152: combination those PCs, edit to ‘combinations of those PCs’

155: add comma after ‘cases’

172 and 231: ‘the ANOISM found’ : grammatically incorrect – the ANOSIM is not capable of finding anything. However, the ANOSIM results showed significant heterogeneity . . .

177: add comma after ‘females’

178 and 201: for all figures, the units on the x and y axes should be indicated

179 and 201: it would be helpful to circle the equivalent symbols to visualize overlap and cited ‘tendencies’ (e.g. line 244: ‘a tendency for data points . . .to concentrate on one side of the cluster’) x and y axes need definition to determine what ‘one side of the cluster’ is referring to.

179 and 201: what do the ‘x’ and ‘y’ axes refer to? This needs to be indicated.

244: What traits explain the ‘tendency’ to concentrate on one side of the cluster?

253: add comma after However

259: add comma after ‘sense’

280: suggest edit to: ‘we conclude that S. typica is a single, well-established species’ to ‘we conclude, based on qualitative and quantitative morphological characters, that S. typica . . .’

301: a prominent papillae ‘a prominent papilla’

Figures (general comments):

·         Unless showing examples of variation across the range of the purported taxon, in redescribing the species it is important that figures are derived from specimens that are as close to Table Bay as possible.

·         For figure lettering it is generally better (as stipulated for most journals) to use sans serif fonts such as Helvitica or Arial rather than Times New Roman.

·         In many figures the scale bars are excessively long.

·         Scale bars should be positioned consistently to the bottom right of each figure and not the center.

·         The weight (thickness) of the lines should be consistent throughout the paper and scale bars should be placed at a consistent distance from the border of each figure (e.g. Figure 11 scale bar is 1 cm from the bottom border; Figure 12 the scale bar is 4 mm from the bottom border and the line is thicker in Figure 12 than Figure 11).

·         Figures need to be centered under the txt and not left justified.

Figure 7: parts a and b cut off in figure. ‘a’ is positioned to close to the left border. (It is generally neater to position figure part labels at an equal distance from the top and the side border i.e.

355: replace & with ‘and’

378: (and throughout the rest of the manuscript) use en-dashes and not hyphens between ranges of numbers and close spaces on either side (i.e. 2.5–18.2% and not 2.5-18.2%; arms I–III p and not arms I – III p.)

391–393: change bold font to normal font

Figure 13: suggest number arms 1–4 on one side of each arm crown

397: (b) change to bold font

406: close spaces and use en dash for 3 – 4, i.e. 3–4

408: ‘sucker loss than genetically’ should read ‘sucker loss rather than genetically’

Figure 14: suggest using finer pen for line drawings, or prepare much larger drawings and reduce size, or SEM images for sucker rims. The lines here are very heavy.

Figure 18: requires reordering of figure parts from left to right, top to bottom:

a              b

c              d

Lettering must not overlap beaks.

Lettering should be placed equidistant from top and side of figure.

Remove text from scale bars.

Position scale bars consistently in relation to the beaks, either bottom left or bottom right.

‘b’ and ‘d’ lettering is cut off at the base

452: The spermatophores are described as simple without any special modifications but it looks from the figure that the cement body might be bipartite (although the figure is not very informative)

453: ‘radula homodont’ should read ‘radula teeth homodont’

453: What is meant by ‘unusual tooth shape’? This needs to be elaborated upon in the Remarks.

Figure 19 (and Figure 14): line drawings are poorly crafted and uninformative. If it is not possible to replace these with better figures, perhaps photomicrograph of the spermatophore cement body and a portion of the radular ribbon, I suggest omitting the figures altogether. However, if the radula has an unusual tooth shape, this should be properly figured (either light micrograph or SEM, or a well-crafted drawing showing more than one row of teeth) as this may be an important taxonomic character.

Figure 20: Lettering not well positioned and too close to left border. Part b: the brightness needs to be increased and contrast decreased.

Figure 21: Lettering not well positioned and bottom of letters are cut off. a and b should be positioned at an equal distance from the corner of each figure part and aligned horizontally (b is much lower than a in this case). The brightness and contrast needs to be adjusted in part a so that the funnel organ can be seen clearly.

471: add space after closing bracket. (in contrast)opaque

Figures 22 and 23 captions: cuttlebone should be one word (check for consistency throughout manuscript).

488–489: The last sentence should be part of the Discussion rather than under the heading ‘Distribution’.

498: systematic status should read ‘taxonomic status’ as this work does not compare this species with others.

500: Suggest reword sentence to read ‘to confirm that based on morphological characters’

523: I am not sure that this could be regarded as the most important finding. Would this not be, instead, the considerable variation in this taxon?

524: Suggest change wording. Instead of ‘further evidence changes of cuttlebones’ revise to ‘This work reinforces that of earlier studies [3,4] that suggest the cuttlebones of small sepiids are particularly susceptible to damage during preservation and long term storage. This has likely led to . . .etc.

545: ‘op’ change to ‘of’

Finally: It would be useful in the Discussion to add a small piece about the biogeography of other benthic invertebrates in the region is if such information is available. Is the distribution recorded for typica one that is shared by other taxa, or does the southern African fauna commonly exhibit an east-west divide?
